# Electrocardiogram Foundation Model Using Temporally Augmented Patient Contrastive Learning

## Abstract

Electrocardiograms ($ECGs$) capture the electrical activity of the heart, offering rich diagnostic and prognostic insights. Traditionally, electrocardiograms are interpreted by human experts, but deep learning is now encroaching on this domain and combining human-like intelligence with machine precision for a deeper insight. Self-supervised pretraining is essential for maximising the potential of scarce medical data. Applied to $ECGs$, patient-contrastive learning has shown promising results, by utilising the natural variations in the cardiac signals. In this study, we introduce **T**emporally **A**ugmented **P**atient **C**ontrastive **L**earning of **R**epresentations ($TA\text{-}PCLR$), a novel approach that incorporates temporal augmentations into a patient contrastive self-supervised foundation model. Trained on one of the largest diverse cohorts of more than six million unlabelled electrocardiograms from three continents, we demonstrate the efficacy of our approach and show its value as a feature extraction tool for small and medium-sized labeled datasets. We also validate the performance on an open-source external cohort, surpassing other pretraining approaches while outperforming an ensemble of fully supervised deep networks on some labels. Additionally, we conduct a detailed exploration of how the pretraining and labeled electrocardiogram dataset distributions impact supervised task performance.

## 1 Introduction

Electrocardiograms ($ECGs$) record cardiac electrical activity as a graph of voltage versus time. Electrodes are placed on the body surface to detect the electrical changes resulting from cardiac muscle depolarization and repolarization. The standard electrode positions define twelve different leads (signals between two specific electrodes) representing cardiac activity along twelve axes, essential for localising the underlying processes. The heart's electrical activity has been recorded since 1887 (Waller, 1887) and has been an important source of information for cardiologists and physicians. Human understanding has since evolved to interpret $ECG$ patterns as manifestations of different health conditions. Being a simple non-invasive investigation, it is part of routine medical care although the expertise for accurate interpretation is not so readily available.

Deep learning has excellent pattern recognition capabilities, surpassing any other methodology, and holds great potential for medical sciences (Esteva et al., 2019). The traditional $ECG$ interpretation depends on the distinct patterns of the waveforms combined with an understanding of the heart function and clinical observations. Human perception is limited by low visual accuracy, gaps in theoretical knowledge, and the complexity of the diverse, non-linear, interrelations (Strodthoff et al., 2021). Deep learning has demonstrated high precision in predicting cardiac diseases (Sau et al., 2023; 2024; Pastika et al., 2024) while opening the possibility of predicting novel labels like age and sex that are beyond human capabilities (Attia et al., 2019). Artificial intelligence has the potential to surpass human capabilities while allowing for accurate diagnosis and risk stratification.

Contrastive learning, a self-supervised pretraining approach, can greatly improve the accuracy for subsequent supervised tasks, especially where the labeled datasets are quite small (Chen et al., 2020). Contrastive learning extracts meaningful representation using a notion of positive and negative instances (Chen et al., 2020). Representations of positive instances are trained to be more similar

and distinct from representations of negative instances. There are diverse contrastive learning approaches, mostly differing in their definitions of the positive and negative instances and the loss computations. In the visual image domain, the positives are usually augmentations (transformations) of the same image while the negatives are augmentations from others. The training retains the meaningful features shared between the positives while discarding the extraneous features.

Contrastive learning has been leveraged for feature extraction from biomedical data with various definitions of data augmentations (Mohsenvand et al., 2020). The augmentations applied to medical data should preserve clinically significant information. Data from the same patient across time is a robust way to encode trivial transformations (Jamaludin et al., 2017; Diamant et al., 2022), thereby allowing the model to reject any artifacts arising from instrument noise, different lead locations, patient movement, etc. Combining different types of random augmentations has remarkably enhanced the performance for contrastive learning (Chen et al., 2020). Incorporating temporal augmentations with patient-based contrastive learning is a key novel feature of our work. We hypothesise that multiple transformations enhance the quality of representations and the selection of clinically insignificant augmentations is essential for the generalisation of the approach. We thus improve on existing patient contrastive methods, by adding temporal augmentation like zero-masking (Soltanieh et al., 2022) and random cropping for our Temporally Augmented Patient Contrastive Learning for $ECG$ Representations ($TA\text{-}PCLR$). In the medical domain, labeled data is often highly scarce, and thus pretraining approaches are essential for maximum utilisation of available data. Foundation models can be pretrained on large unlabeled datasets, to learn general features that can be leveraged for a range of subsequent supervised tasks (Zhang & Metaxas, 2024). We train a foundation model on $6,174,025$ $ECGs$ to improve generalisation for $ECG$ feature extraction and explore the effect of pretraining data on the quality of the learned representations. The main contribution of our work is to present a new foundation model for $ECG$ interpretation. Our model achieves state-of-the-art results due to:

- a new method: Temporally Augmented Patient Contrastive learning that incorporates augmentations along the temporal axis.
- a new multi-center dataset we constructed for training with over six million $ECGs$ with four cohorts from three different continents.

Our augmentation techniques improve on previous contrastive learning techniques based on contrasting between exams from the same patient (Diamant et al., 2022). Our newly constructed dataset allows us to showcase, the effect of training data demographics, on the quality of the learned representations. We present a foundation model that outperforms other, much larger, self-supervised foundation models (Song et al., 2024) and is close to highly optimised fully supervised benchmarks (Strodthoff et al., 2021).

Section 2 presents a brief overview of the past efforts conducted for the contrastive learning of electrocardiograms. Section 3 describes the implementation details of our approach, followed by the performance analysis in Section 4. Section 5 concludes with the highlights of our research findings and presents some suggestions for future research.

## 2 LITERATURE BACKGROUND

**Contrastive learning**  Contrastive learning has shown remarkable improvement for image classification tasks, by incorporating a self-supervised pretraining (Hadsell et al., 2006). In the computer vision domain, pretraining enhances the similarity of the images sharing context (positives) while reducing that from distinct images (negatives). Efficient techniques such as $InfoNCE$ (van den Oord et al., 2018) present a robust encoding of the contrastive loss, and the $SimCLR$ (Chen et al., 2020) introduces the idea of a non-linear projection layer for performance improvement.

**Data augmentations**  Data augmentations are essential for contrastive learning to enhance meaningful context and reject spurious information. The augmentations of visual images can be performed by using different segments, views, and coloring of the original image, retaining the meaningful context (Chen et al., 2020). The dimensions for augmentation are more limited for the time series data and mainly involve noise addition, time-masking, cropping, shuffling, inverting, etc (Wen et al., 2021). Previous work has shown that good augmentations are crucial to pre-

serving only meaningful features and greatly affect the generalisation for subsequent supervised tasks. The diversity in negative and positive examples is also essential for learning meaningful features. Combining multiple augmentations has been proved to reinforce the learning process (Chen et al., 2020; Gopal et al., 2021). For medical data, augmentations are also restricted by possible clinical implications, therefore synthetic augmentations have to be taken with care. Past work has shown that while some augmentations might improve the performance on one task, they may have adverse effects on others (Lee et al., 2022; Raghu et al., 2022).

**Contrastive learning for** $ECG$**s** Contrastive learning has been employed for $ECG$ representation learning. $CLOCS$ (Kiyasseh et al., 2021) presents the idea of using $ECG$ from the same patient as a meaningful context, with the different leads and non-overlapping slices from the same $ECG$ as positives, thereby incorporating multiple positive $ECG$s in the batch, thus improving performance over the $SimCLR$ baseline (Chen et al., 2020). $PCLR$ (Diamant et al., 2022) takes the concept further defining $ECG$ from the same patient over time as positive instances and demonstrates superior performance compared to previous approaches. The contrastive heartbeats ($CT$-$HB$) (Wei et al., 2022) splits the individual heartbeats from an $ECG$ recording and defines heartbeats from the same $ECG$ as positives and implements a variant of triplet loss (Wang et al., 2019). Soltanieh et al. (2022) systematically explores a spectrum of time series augmentations for $ECGs$ including time-warping, permutation (slice and shuffle), inverting, and scaling, which nonetheless could have clinical implications. Physiologically-inspired spatial and temporal augmentations including axis rotation, scaling, and zero-masking, are combined by the $3KG$ (Gopal et al., 2021) for self-supervised pretraining, improving $ECG$ classification performance. $ECG-FM$ (McKeen et al., 2024) employs a multi-layer convolutional feature extractor and a transformer encoder for feature extraction with random-lead-masking augmentation. The joint cross-dimensional contrastive learning approach (Liu et al., 2023) is based on learning $ECG$ representation by contrasting $ECG$ signals against images incorporating several modes of augmentations. $MERL$ (Liu et al., 2024) contrasts $ECGs$ with clinical reports to provide the possibility of zero-shot inference.

**Generative pretraining:** Self-supervised pretraining has been implemented following generative approaches, such as masked autoencoders ($MAE$) reconstructing random masked $ECG$ segments (Gedon et al., 2021; Na et al., 2024). Hybrid techniques combining contrastive learning and generative pretraining based on transformer architecture have been implemented for $ECG$ feature extraction (Song et al., 2024).

**Foundation models** Foundation models are defined by the flexibility to facilitate generic downstream tasks by exploiting huge pre-training cohorts (Zhang & Metaxas, 2024). All of the self-pretraining methodologies have the potential to adapt to any generic task and large cohorts can further enhance the model capabilities. HeartBeiT (Vaid et al., 2023) exploits vision transformer architecture to present an $ECG$-based foundational model. (Song et al., 2024) trained a foundation $ECG$ model on more than a million $ECGs$ exploiting a hybrid approach combining contrastive learning with generative pretraining involving vision transformers. $ECG-FM$ (McKeen et al., 2024) employs $wav2vec$ architecture with a convolutional feature extractor and a BERT-like transformer encoder trained on 1.6 million $ECGs$. Foundation models have also been developed following a supervised approach (Li et al., 2024) where the generalization capabilities may be limited.

**Our approach** Our work is a natural extension and improvement of these efforts. We combine patient-based augmentation with simple temporal augmentations based on random zero-masking and cropping without impacting the underlying clinical information, thus making the approach robust. We further train on a large diverse cohort for improved generalization and explore the impact of the pretraining data demographics on the learned representations. We also demonstrate that label-based performance comparisons for diverse datasets may not reflect the true merits of an approach.

## 3 MATERIALS AND METHODS

### 3.1 COHORTS

The study employs a range of large, diverse $ECG$ cohorts from three continents: Beth Israel Deaconess Medical Center ($BIDMC$) (Pastika et al., 2024) from the United States, Clinical Out-

comes in Digital Electrocardiography ($CODE$) (Ribeiro et al., 2019) from Brazil, Shanghai Zhong-shan Hospital cohort dataset ($SHZS$) from China, Vanderbilt University Medical Center cohort ($VUMC$) (Aras et al., 2023) from United States, UK Biobank ($UKB$) (Sudlow et al., 2015) from United Kingdom and Physikalisch-Technische Bundesanstalt ($PTB$-$XL$) dataset (Wagner et al., 2020) from Germany. Table 1 presents the data used in current research in terms of the number of unique patients with more than one $ECG$ and the corresponding number of $ECGs$ for the contrastive learning pretraining cohorts: $BIDMC$, $CODE$, $VUMC$, and $SHZS$, and the total $ECGs$ for datasets used in performance validation: $UKB$ and $PTB$-$XL$. We denote the combined pre-training cohort as $BCSV$ consisting of more than six million individual $ECGs$, with each $ECG$ comprising eight leads. Appendix A provides important information about the dataset demographics in Table 6, while further details can be obtained from the corresponding references.

Table 1: Datasets

| No. | Cohorts | [1] Patients* | ECGs |
|---|---|---|---|
| 1 | $BIDMC$ (United States) (Pastika et al., 2024) | $127,041$ | $1,106,886$ |
| 2 | $CODE$ (Brazil) (Ribeiro et al., 2019) | $424,577$ | $1,123,903$ |
| 3 | $SHZS$ (China) | $420,957$ | $2,257,485$ |
| 4 | $VUMC$ (United States) (Aras et al., 2023) | $252,306$ | $1,685,737$ |
| 5 | $BIDMC$+$CODE$+$SHZS$+$VUMC$ ($BCSV$) | $1,224,881$ | $6,174,011$ |
| 6 | $UKB$ (United Kingdom) (Sudlow et al., 2015) | - | $70,655$ |
| 7 | $PTB$-$XL$ (Germany) (Wagner et al., 2020) | - | $21,800$ |

*Unique patients with more than one $ECG$

## 3.2 CONTRASTIVE LOSS

The contrastive loss employed for the current work is the $InfoNCE$ loss (van den Oord et al., 2018) applied to the non-linear projections similar to $SimCLR$ (Chen et al., 2020). Given that $z_i$ and $z_j$ are the non-linear projections of representations from two different augmented $ECGs$ belonging to the same patient, the similarity between $z_i$ and $z_j$ is enhanced over all other instances in the batch by applying a $softmax$ (Bridle, 1989) over the similarity values. The loss function then implements the following equation 1 where $\tau$ is a temperature coefficient defining how soft or hard the $softmax$ constrains the similarity distributions and $N$ denotes the number of pairs in the batch. The $\mathbb{I}_{[k \neq i]} \in 0, 1$ is an indicator function evaluating to 1 only if $k \neq i$.

$$\ell_{i,j} = - \log \frac{\exp(\text{sim}(z_i, z_j)/\tau)}{\sum_{k=1}^{2N} \mathbb{I}_{[k \neq i]} \exp(\text{sim}(z_i, z_k)/\tau)} \tag{1}$$

Where sim is the cosine similarity defined as:

$$\text{sim}(z_i, z_j) = \frac{z_i^\top \cdot z_j}{||z_i|| \cdot ||z_j||} \tag{2}$$

## 3.3 AUGMENTATION

Generalization is an important aspect of a foundation model, we refrain from scaling, rotating, or frequency-warping employed in some past pre-training approaches (Soltanieh et al., 2022) that may potentially impact the performance for any unforeseen future supervised tasks where the scale, axis, or rhythm may be important information (Raghu et al., 2022). We reason that contrasting for patient identity (Diamant et al., 2022) is a natural way of encoding trivial transformation and we combine it with temporal augmentations, such as zero-masking (Gopal et al., 2021; Soltanieh et al., 2022)

and random cropping thus ensuring that the pretraining will be relevant for any potential future supervised task.

We define the positive views as augmentations of random slices from different $ECGs$ of the same patient. The number of $ECGs$ from each patient greatly differs thus the training epoch is defined as one complete iteration for all unique patients with the positive views randomly sampled at training time. In this way, the training is not biased by patients having more $ECGs$, while exploiting the available data diversity. In contrast to Diamant et al. (2022), $ECGs$ in a positive pair are always unique instances. The input window size of seven seconds allows random cropping ($RC$) by using a different patch from the same ten-second $ECG$ for each epoch. The additional temporal augmentation includes zero-masking (Soltanieh et al., 2022; Raghu et al., 2022) that is unimpactful of any intrinsic clinical information. The transformations are applied at the training time so for the same $ECG$, the slices and masks are unique for each epoch, increasing the diversity of the positive samples. We experimented with applying zero-masking to the same random segments for each lead ($RZM$), different random segments for each lead ($RLZM$), masking random leads ($RLM$) (Oh et al., 2022), and a novel notion of using the raw and filtered ECGs as augmentations ($RF$)[1]. We retained the simpler configuration of $RZM$, which showed the best performance.

### 3.4 PREPROCESSING

The standard procedure for $ECG$ recording involves measurements from 12 leads recorded for 10 seconds with sampling rates typically at 400 to 500 samples per second ($Hz$). For model development, we used eight $ECG$ leads as four leads are linear combinations of other leads and thus do not impart additional information (Eem et al., 2020). We apply a bandpass filter (0.5 to 100 $Hz$) and a notch filter relevant to the mains frequency and interpolate $ECGs$ from different sources to a standard sampling frequency of 400 $Hz$. We retain the original scale of the $ECGs$ in millivolts. The final input shape to the contrastive learning model is $2800 \times 8$ (7 second signal).

### 3.5 ARCHITECTURE

Figure 1 presents an overview of the $TA\text{-}PCLR$ architecture. The $ECGs$ for the same patient are treated as positive views while all other $ECGs$ in the batch are negative views. For a fair comparison, we use the backbone architecture from Ribeiro et al. (2020) with non-linear projections (similar to $PCLR$ (Diamant et al., 2022)). The contrastive loss is applied to the non-linear projections of the $ECGs$. The output of the model is 256 features or embeddings learned from the $ECGs$ that can be exploited for any downstream supervised training. The model is implemented in Tensorflow (Abadi et al., 2015) (2.10.1). The total number of parameters is less than 6 million and thus the model is much more compact, as compared to the transformer-based architectures like $ECG - FM$, having more than 300 million parameters (McKeen et al., 2024).

### 3.6 TRAINING

The contrastive loss performs best with larger batches due to the larger variation of the negative instances (Chen et al., 2020), but the computation resources limit the batch size. We use a batch size of 1024 (512 patients) and train the model for 200 epochs using the Adam optimizer (Kingma & Ba, 2014). The initial learning rate is 0.1 and then decayed according to a half-period cosine schedule (Loshchilov & Hutter, 2016), similar to previous approaches (Chen et al., 2020; Diamant et al., 2022). The training time in minutes per epoch on NVIDIA GeForce RTX 3090 is approximately 5 for the $BIDMC$ dataset and 50 for the $BCSV$ dataset. This is notably shorter than comparative approaches in the literature taking weeks on multi-gpu configurations (Cheng et al., 2021; Song et al., 2024; McKeen et al., 2024).

---

[1] For $RZM$ and $RLZM$ a 20% segment is masked while a 10% masking probability is applied for $RLM$. The $RF$ configurations contrast raw vs. raw with 40%, filtered vs. filtered with 40%, and raw vs. filtered with 40% probability.

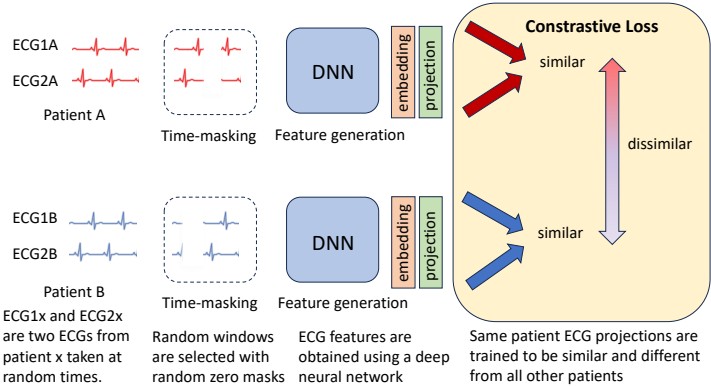

Figure 1: The $TA\text{-}PCLR$ pretraining overview. Patient A has $ECG1A$ and $ECG2A$ while Patient B is another patient in the same batch with $ECG1B$ and $ECG2B$. The $ECGs$ are transformed by temporal augmentations and converted by the network to a 256-feature vector. The contrastive loss brings the projections from the same patient $ECGs$ closer and apart from others in the batch.

## 4 PERFORMANCE ANALYSIS

The $TA\text{-}PCLR$ is a pretraining paradigm for unlabelled data, thus the performance of the learned embeddings is evaluated by subsequent supervised training similar to previous works (Chen et al., 2020). Following are the different performance evaluation configurations employed in the current work, depending on the experiment at hand:

- Linear evaluation: A linear probe is a standard methodology to compare the expressiveness of the features learned during unsupervised pretraining approaches (Chen et al., 2020). The feature-generating model is frozen while a single neuron is trained to predict each label.

- Multi-layer perceptron ($MLP$): We also train a two-layer $MLP$ for the supervised task to allow the model to learn a non-linear mapping of the $TA\text{-}PCLR$ generated features.

- Fine-tuning: After training a linear model the feature-generating model can also be allowed to update and further improve the performance.

Validation is performed along two main dimensions: proof of concept by internal validation of the approach for different experimental configurations, and external validation by comparing to previous benchmarks. The test evaluation involves a single prediction from a random crop of each test $ECG$. We mainly report macro-averaged area under the receiver-operating characteristic curve $AUROC$ or $AUC$ as a threshold-free assessment of classification performance and mean absolute error $MAE$ in years for the age regression performance, as suggested previously (Wagner et al., 2020; Strodthoff et al., 2021). Unless otherwise specified, we report the mean of ten independent runs for each task.

### 4.1 PROOF OF CONCEPT

**Experimental setup** The $BIDMC$ dataset is primarily employed for model development and we demonstrate the superiority of our pretraining paradigm on this cohort. The supervised tasks for $BIDMC$ involve predicting an individual's age, sex, and probability of five-year mortality from an $ECG$. The self-supervised pretraining is performed on the $ECGs$ of the patients having more than two $ECGs$ (details in Table 1) with $80\%$ of the patients included in the training and the rest in the validation set. The supervised tasks are then implemented using a train, validation, and test split of $50\%$, $10\%$, and $40\%$ for $1,169,387$ labeled $ECGs$. The $TA\text{-}PCLR$ supervised tasks in this section use an $MLP$ for the label prediction (details in Appendix D). The extracted features are standardized and the learning rate is manually optimised within the range of $[0.00001, 0.01]$.

Table 2: Ablation study exploring the contribution from each component of our approach on the supervised task for $BIDMC$ age, sex, and five-year mortality predictions. All the implementations use patient contrastive learning and random cropping ($RC$). The additional augmentations tested include random lead zero masking ($RLZ$), raw-filtered ($RF$), random lead masking ($RLM$), and random zero masking ($RZ$). The best performance is indicated in bold while the second best is underlined.

| No. | Augmentation | Pretraining cohort | Age ($MAE$) | Sex ($AUC$) | Mortality ($AUC$) |
|-----|-------------|--------------------|-------------|-------------|-------------------|
| 1 | Patient-based | | 8.3061 | 0.9145 | 0.7729 |
| 2 | Patient + $RC$ + $RLZM$ | BIDMC | 7.8860 | 0.9338 | 0.7864 |
| 3 | Patient + $RC$ + $RZM$ | | 7.8498 | 0.9351 | 0.7883 |
| 4 | Patient + $RC$ + $RZM$ + $RF$ | | 7.9018 | 0.9335 | 0.7873 |
| 5 | Patient + $RC$ + $RZM$ + $RLM$ | | 7.9353 | 0.9318 | 0.7900 |
| 6 | Patient + $RC$ + $RZM$ | $BCSV$ | **7.7849** | **0.9393** | **0.7926** |

**Ablation study** Our approach consists of patient-based contrastive learning (Diamant et al., 2022) and temporal augmentations, together with our unique diverse dataset. Table 2 represents an ablation study highlighting the contribution of each component. The top row shows the performance of the patient-based contrastive learning ($PCLR$) pretrained on the $BIDMC$. The combined patient contrastive and temporal augmentations significantly improve the performance. We experiment with several augmentations including random cropping and zero-masking, as well as a novel raw-filtered augmentation with $RZM$ resulting in the best performance. Finally, training with the combined $BCSV$ data further enhances the performance, providing 6.27%, 2.71%, and 2.55% improvement for age, sex, and mortality, respectively. Hence, we demonstrate that temporal augmentations (on top of $PCLR$) enhance the model performance and an increased dataset size further improves task-specific efficacy.

**How performance compares to supervised training for different train data sizes?** Figure 2 compares the performance of the $TA\text{-}PCLR$ pretraining versus a randomly initialized $ResNet$ network with a similar backbone used previously for $ECG$ classification (Ribeiro et al., 2020). The test is conducted on a frozen feature-generating backbone with an $MLP$ head, for different sizes of the labeled training data. The $TA\text{-}PCLR$ performance is reported as a mean of ten independent runs while the $ResNet$ is trained for a single run. The $TA\text{-}PCLR$ outperforms the $ResNet$ for smaller training data sizes up to 200 k when it converges. For age, sex, and mortality label prediction with 1000 training samples, the performance improvement is 15.84%, 16.94%, and 8.18%, respectively. It should be noted that $TA\text{-}PCLR$ performance can be further improved by fine-tuning. We thus demonstrate that the performance of $TA\text{-}PCLR$ is superior to fully-supervised $ResNet$ specifically for smaller datasets, which are often prevalent in the medical domain.

## 4.2 MULTIPLE PRETRAINING COHORTS COMPARISON

An important aspect of the work is to explore how training can benefit from the huge corpus of available $ECG$ data in terms of data size and diversity. Prior work hints that the results for a supervised task also depend on the underlying distribution of the dataset. For example, age and sex prediction shows higher performance for healthy subjects compared to unhealthy (Strodthoff et al., 2021). The cardiac signal has been known to have ethnicity signatures (Mansi & Nash, 2004), thus the ethnic distribution of the training cohort can also affect the performance.

**Experimental setup** The following experiment is designed to study the effect of data demographics on the performance of pretraining approaches. Pretraining is accomplished on the four cohorts

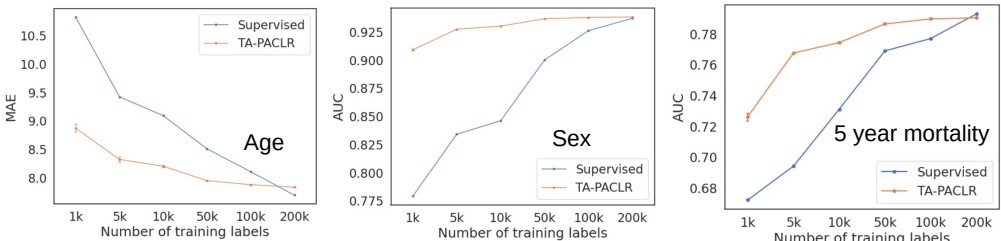

Figure 2: Comparison of $TA\text{-}PCLR$ (orange) with supervised learning (blue) for labels from the BIDMC dataset shows the remarkable improvement achieved through the proposed approach. Left) Age regression; center) Sex classification; right) Five-year mortality prediction.

$BIDMC$ (USA), $CODE$ (Brazil), $SHZS$ (China), and $VUMC$ (USA), having different ethnic distributions, patient count, and $ECG$ numbers. The $BIDMC$, $CODE$, $SHZS$, $PTB\text{-}XL$, and $UKB$ datasets contain the age and sex of the patients for each $ECG$. The pretrained models are used to extract features from these cohorts. The features are then leveraged for age and sex prediction with the train/val/test splits for all datasets consisting of $10k/2k/2k$ labeled instances, to remove any dataset size bias from the supervised training. Figure 3 compares the performance of the age and sex prediction across multiple pretrained feature extraction models and labeled datasets, using a similar configuration for the supervised setup. The right-hand panel presents the $AUC$ for the sex prediction while the left-hand panel shows the $MAE$ for age prediction, as the mean of six runs.

**How does the pretraining cohort affect performance?** The following observations and insights can be obtained from the test that we believe to be essential for future work. The model pretrained on $BCSV$ (i.e. the combination of $BIDMC$, $CODE$, $SHZS$, and $VUMC$ datasets) outperforms for all labels and has the best generalisation capabilities for a foundation model. Apart from the $BCSV$, the model pretrained on the same labeled dataset generally performs the best, thus highlighting the importance of external open-source cohorts for performance comparison. It is interesting to note, that while the $CODE$ and $SHZS$ have more patients and $ECG$s, the performance is generally decreased compared to the secondary care cohorts of $BIDMC$ and $VUMC$. A plausible explanation can be that the learned features are more expressive when pretrained from a more diseased population as there is more diversity in $ECG$ patterns, which is the case for the $BIDMC$ and $VUMC$ datasets. Moreover, the performance does not only depend on the number of unique patients but also the diversity of the positive examples i.e., the $ECGs$ per patient. Looking back at Table 1, the $BIDMC$ has far fewer unique patients and $ECG$s than $VUMC$, but has a higher number of $ECGs$ per patient that may help it achieve comparable performance.

**Does the labeled dataset impact performance?** The performance metric for a labeled dataset also depends on the distribution of the underlying population health. The $UKB$ supervised tasks show the highest $AUC$ and lowest $MAE$ for the same pretrained model, being a healthy volunteer cohort. Therefore, a metric for a particular label cannot be used for performance comparison across diverse datasets. Several previous works compare their results based on a specific training task (McKeen et al., 2024), which may be meaningful only when model evaluation is compared on the same dataset and ideally the same train/test splits. The model, pretrained on the $CODE$ and $SHZS$ datasets, shows lower performance compared to much smaller datasets. One reason can be that these datasets come from a healthier population and consequently reduced diversity. Additionally, the population of $SHZS$ being more ethnically distinct can also impact the performance on diverse datasets. The performance of the patient-based approach is highly dependent on the diversity, number, and health of the underlying population.

### 4.3 EXTERNAL VALIDATION

Standard benchmarks for computer vision approaches based on open source datasets like Imagenet (Deng et al., 2009), CIFAR10 (Krizhevsky, 2009), and COCO (Lin et al., 2014) have greatly facilitated unbiased performance comparison. Section 4.2 exploring the effect of pretraining co-

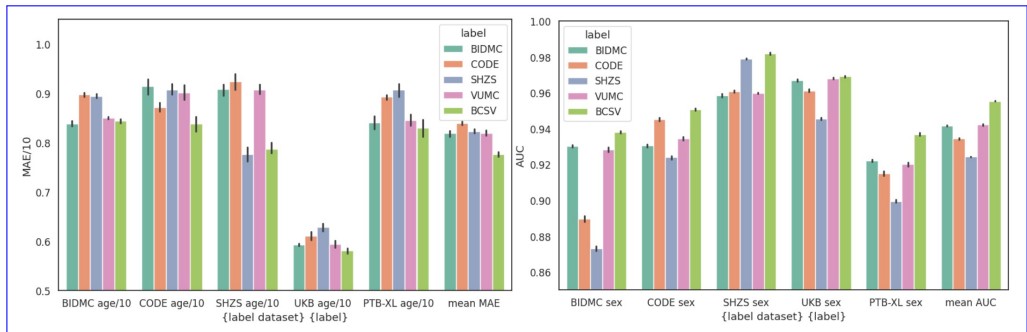

Figure 3: Performance comparison for the different pretraining and labeled datasets: right) Sex prediction, left) Age prediction. The different colors indicate different pretraining cohorts. The x-axis denotes the label and the labeled dataset while the y-axis denotes the metric under observation. The performance for $BCSV$ is the best across all labels and datasets.

horts, shows that the performance can vary due to data demographics, thus a fair comparison can be established by exploitation of open-source cohorts that can be easily accessible for future works.

**Experimental setup**   We use the open-source $ECG$ dataset $PTB$-$XL$ (Wagner et al., 2020) for external validation. The dataset is well explored in the literature and provides a range of benchmarks for performance comparison including fully supervised and pretraining strategies. The comparison here is limited to published results from literature where the authors may have fully optimised all aspects of their approach. The values for all metrics may not be available as indicated by '-' in the tables. Appendix B provides a more detailed comparison with other pretraining approaches in Table 7 and detailed classification metrics in Table 8. The model is pretrained on the $BCSV$ cohort while the supervised setup consists of a linear classification head. Detailed hyperparameters optimisation is not performed; only learning rates are optimised from a range of [0.00001, 0.1]. A simple fine-tuning procedure is implemented without exploiting advanced techniques. The details of the experimental hyperparameters are provided in Appendix

D. We explore two levels of diagnostic labels from $PTB$-$XL$ denoting cardiac abnormalities: Sub-classes refer to 23 morphological labels, and Super-classes are 5 overarching diagnostic classes. The dataset is divided into ten stratified folds, with fold 10 suggested for the test, fold 9 for the validation, and the remaining folds for the training (as per Wagner et al. (2020)). Further details about the dataset distribution and the labels can be obtained from Wagner et al. (2020).

**How the TA-PCLR compares with supervised models?**   Table 3 compares the results with previously published benchmarks for the $PTB$-$XL$ employing fully supervised training. The training for super and sub classes is accomplished as multi-label classification where an instance can belong to more than one class, thus we report macro $AUC$. We compare with the best-performing benchmark from a study exploring deep neural networks (Strodthoff et al., 2021), which combines the predictions generated for different slices of the $ECG$ signal, with a highly optimised ensemble of state-of-the-art deep neural networks. Bickmann et al. (2024) implements fully supervised learning for the prediction of $PTB$-$XL$ super-classes, utilizing advanced deep learning architecture (InceptionTime (Ismail Fawaz et al., 2020)). The $TA$-$PCLR$ with the linear probe has slightly lower performance than the fully supervised approaches as the feature extraction backbone is frozen. The performance improves significantly when the model is further fine-tuned allowing all weights to update. The age and sex predictions outperform the aggregated predictions from the ensemble network while diagnostic classes surpass the InceptionTime network pereformance (Bickmann et al., 2024). We set new benchmarks for age and sex prediction with 5.2% and 1.9% improvement, respectively.

**How does the TA-PCLR compare with other pretraining methodologies?**   We now compare the $TA$-$PCLR$ approach to several representative pretraining approaches in Table 4. The linear classification head is trained individually for each label in super-classes: Myocardial Infarction ($MI$), ST/T change ($STTC$), Conduction Disturbance ($CD$), and Hypertrophy ($HYP$). We compare with a state-of-the-art masked autoencoder-based foundational model by Song et al. (2024) and a

Table 3: Performance Comparison with Supervised training for $PTBXL$: Age, sex, diagnosis super/sub classes

| No. | Method | MAE | | AUC | |
| | | Age | Sex | Super-classes | Sub-classes |
|---|---|---|---|---|---|
| 1 | Strodthoff et al. (2021) (ensemble) | 7.12 | 0.928 | **0.934** | **0.933** |
| 2 | Bickmann et al. (2024) (Inception) | - | - | 0.902 | - |
| 3 | $TA$-$PCLR$ linear | 7.57 | 0.938 | 0.889 | 0.919 |
| 4 | $TA$-$PCLR$ fine-tuning | **6.75** | **0.946** | 0.919 | 0.928 |

Table 4: Comparing with pretraining approaches for the PTB-XL Super-classes: Myocardial Infarction ($MI$), ST/T change ($STTC$), Conduction Disturbance ($CD$), and Hypertrophy ($HYP$).

| No. | Method | $MI$ | $STTC$ | $CD$ | $HYP$ | $NORM$ | Mean |
|---|---|---|---|---|---|---|---|
| 1 | Song et al. (2024) | 0.8318 | 0.8165 | 0.8411 | 0.8135 | - | - |
| 2 | Liu et al. (2023) | - | - | - | - | - | 0.8648 |
| 3 | $TA$-$PCLR$ linear | **0.8948** | **0.8848** | **0.8885** | **0.8669** | **0.9173** | **0.8905** |

cross-dimensional approach Liu et al. (2023) (reporting a superior performance as compared to the $3KG$ (Gopal et al., 2021), and $PCLR$), although noting that they removed instances with multiple labels to implement the task as multi-class. We present their result while noting that the data will be a subset of the dataset with a simpler task. The performance of $TA$-$PCLR$ with linear probe significantly outperforms these more complex approaches with an improvement of 5 to 8 percent over Song et al. (2024) for individual classes and 2.97% above Liu et al. (2023). Additional comparisons in Appendix B Table 7 further substantiate the superiority of the proposed approach while Table 8 provide more detailed metrics for the the results presented in this section.

## 5 CONCLUSIONS AND FUTURE WORK

In this work we present $TA$-$PCLR$, applying a novel combination of temporal augmentations and patient-based $ECG$ contrastive learning, enhancing performance in downstream supervised tasks. We demonstrate that $TA$-$PCLR$ is superior to fully supervised training methods on small to medium-sized datasets, proving to be especially valuable in scenarios where labeled data is limited.

Pretraining with combined datasets from three continents, forming one of the largest and most diverse, multi-site $ECG$ cohorts, further improves the performance for downstream supervised tasks and sets new benchmarks when evaluated in external validation using the $PTB - XL$ dataset. Looking ahead, we plan to explore the impact of additional data augmentations on the learned representations and focus on enhancing the interpretability of the features learned by $TA$-$PCLR$ (some work on interpretability is presented in Appendix C). The current work demonstrates the capabilities of our foundation model for a range of generic tasks without increasing the model's complexity. An interesting research direction for future work is exploring model scalability while noting that the rule of ten times more data than model parameters from prior research (Alwosheel et al., 2018) and our remarkable results attest to the fact that the model size maybe adequate. The strong performance, generalizability, and efficiency of $TA$-$PCLR$ in terms of training time and network size, positions it as a powerful foundation model.

# 6 ETHICS STATEMENT

Our research complies with all relevant ethical regulations and details of the ethics approval are provided in Table 5.

Table 5: Datasets

| No. | Cohorts | Ethics Approval |
| --- | --- | --- |
| 1 | $BIDMC$ | Beth Israel Deaconess Medical Center Committee on Clinical Investigations (IRB protocol # 2023P000042). |
| 2 | $CODE$ | Research Ethics Committee of the Universidade Federal de Minas Gerais (protocol 49368496317.7.0000.5149) |
| 3 | $SHZS$ | Institutional Research Board of Zhongshan Hospital (No. B2023-253R) with a waiver of patient consent |
| 4 | $VUMC$ | The Vanderbilt component of this study was reviewed and approved by the Institutional Review Board (#212147) |
| 5 | $UKB$ | North West Multi-Centre Research Ethics Committee application ID 48666 |
| 6 | $PTB\text{-}XL$ | The Institutional Ethics Committee approved the publication of the anonymous data in an open-access database (PTB-2020-1). |

# 7 REPRODUCIBILITY STATEMENT

The pretraining data consist of private cohorts thus the trained network cannot be released but the details provided in Section 3 are sufficient to reproduce the methodology.

AUTHOR CONTRIBUTIONS

ACKNOWLEDGMENTS

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

## A  DATASET ANALYSIS

Table 6 presents the demographics of the population included in the study. Further details can be obtained from the corresponding references. Although the ethnicity information for most cohorts is not available, depending on the geographical location the predominant ethnicity can be inferred.

## B  ADDITIONAL RESULTS FOR PTB-XL

Table 7 compares the macro $AUC$ values reported by Liu et al. (2024) for state-of-the-art pretraining approaches, with our $TA\text{-}PCLR$ approach. We repeat the tests for ten independent runs with 1%, 10%, and 100% random splits of the training data while the validation and test splits remain the same. The learning rates are optimized from the range of $[0.0001, 0.01]$. The parameters for 100% split are provided in Table 9, while learning rates used for 10% split are 0.001 for the sub classes and 0.01 for the super classes. Similarly, the learning rates for 1% split are 0.05 for sub classes and 0.01 for the super classes. It should be noted here that $MERL$ has some auxiliary supervision during training, in the form of diagnostic ecg-report alignment that enables the model to perform zero-shot prediction. The $TA\text{-}PCLR$ outperforms for all except for the 1% split for the super classes, where it has the second-best performance.

Table 8 provides detailed metrics for the classification tasks performed in Tables 3 and 4. The results are obtained from ten random runs and presented as macro $AUC$ mean with 95% confidence interval. The metrics like precision, recall, $F1$, and accuracy greatly depend on the threshold for the binary output. A threshold of 0.5 is used for the reported results but the metrics will be significantly improved by optimizing the threshold.

## C  INTERPRETABILITY

The t-SNE (t-distributed Stochastic Neighbor Embedding) (van der Maaten & Hinton, 2008) is a non-linear dimensionality reduction algorithm that can help to visualize high-dimensional data in two or three dimensions. The pretraining is not supervised so it encodes the generic $ECG$ features.

Table 6: Baseline characteristics of the population used in the current research.

|  | $BIDMC$ | $CODE$ | $SHZS$ | $VUMC$ | $UKB$ | $PTB\text{-}XL$ |
|---|---|---|---|---|---|---|
| Location | United States | Brazil | China | United States | United Kingdom | Germany |
| Patients* | $127,041$ | $424,577$ | $420,956$ | $252,306$ | $66,402$ | $18,869$ |
| ECGs | $1,106,886$ | $1,123,903$ | $1,560,551$ | $1,412,012$ | $70,655$ | $21,799$ |
| Age mean | $57.99$ | $56.00$ | $52.08$ | $52.08$ | $65.35$ | $62.36$ |
| Age IQR | $23.02$ | $23.00$ | $27.00$ | $27.00$ | $12.00$ | $23.00$ |
| Male | $63,006$ | $165,285$ | $233,808$ | $233,808$ | $32,191$ | $9,640$ |
| Female | $640,35$ | $259,292$ | $187,148$ | $187,148$ | $34,211$ | $9,229$ |
| Hispanic | $7,077$ | - | - | - | - | - |
| White | $84,265$ | - | - | - | - | - |
| Black | $17,778$ | - | - | - | - | - |
| Asian | $5,315$ | - | - | - | - | - |
| Other | $12,606$ | - | - | - | - | - |
| Mortality[!] | $21\%$ | - | - | - | - | - |

* Patients with more than one ECG.
[!] Five-year mortality.

Figure 4 right panel shows the correlations between the different features (features are standardized and the features with zero standard deviations are removed from the further study). The correlations show that features are not very correlated and thus more expressive. The left panel shows the two principle components obtained by the t-SNE, with five super classes in different colors. The classes are multi-label and not mutually exclusive thus mostly expressed as gradients instead of clustering. The classes are also composite so the same classes can be observed to be located in the different but nearby regions of the embedding space.

Figure 5 shows principle t-SNE components for other labels like sex, NORM, and age. The components are obtained for features with a correlation greater than $0.3$ with the corresponding label. The plots for sex and NORM show that the embedding space can separate the genders and normal versus abnormal in opposite directions. The right panel shows the different age groups show a gradient in the t-SNE representation.

Figure 6 further explores the interpretability of the model using $GradCam$ (Selvaraju et al., 2017) . The top two $ECG$s are examples with positive class $STTC$, while the bottom two $ECG$s are negatives. The gradients are superimposed on the input signal and are represented in red color where the darker color indicates higher importance. The $STTC$ class is related to abnormalities in the ST segment (de Luna et al., 2005) of the ECG. The plots show that the model can recognize the normal samples using the region around the ST segments probably by learning the specific shape in a normal person. The positive samples with abnormalities do not give any importance to this region.

Table 7: Linear probe result comparison with SOTA approaches for the PTB-XL Super and sub classes classification.

| No. | Method | Super-classes | | | Sub-classes | | |
|---|---|---|---|---|---|---|---|
| | | 1% | 10% | 100% | 1% | 10% | 100% |
| 1 | SimCLR Chen et al. (2020) | 0.634 | 0.698 | 0.735 | 0.608 | 0.683 | 0.734 |
| 2 | BYOL Grill et al. (2020) | 0.717 | 0.738 | 0.764 | 0.572 | 0.674 | 0.716 |
| 3 | BarlowTwins Zbontar et al. (2021) | 0.729 | 0.760 | 0.784 | 0.626 | 0.708 | 0.743 |
| 4 | MoCo-v3 Ci et al. (2022) | 0.732 | 0.766 | 0.783 | 0.559 | 0.692 | 0.767 |
| 5 | SimSiam Chen & He (2021) | 0.731 | 0.727 | 0.756 | 0.625 | 0.693 | 0.764 |
| 6 | TS-TCC Eldele et al. (2023) | 0.707 | 0.759 | 0.789 | 0.535 | 0.670 | 0.779 |
| 7 | CLOCS Kiyasseh et al. (2021) | 0.689 | 0.734 | 0.763 | 0.579 | 0.725 | 0.762 |
| 8 | ASTCL Wang et al. (2024) | 0.725 | 0.773 | 0.810 | 0.619 | 0.688 | 0.765 |
| 9 | CRT Zhang et al. (2024) | 0.697 | 0.782 | 0.772 | 0.620 | 0.708 | 0.787 |
| 10 | ST-MEM Na et al. (2024) | 0.611 | 0.669 | 0.713 | 0.541 | 0.579 | 0.636 |
| 11 | MERL Liu et al. (2024) | **0.824** | 0.862 | 0.887 | 0.649 | 0.806 | 0.847 |
| 12 | *TA-PCLR* | 0.788 | **0.870** | **0.889** | **0.685** | **0.849** | **0.918** |

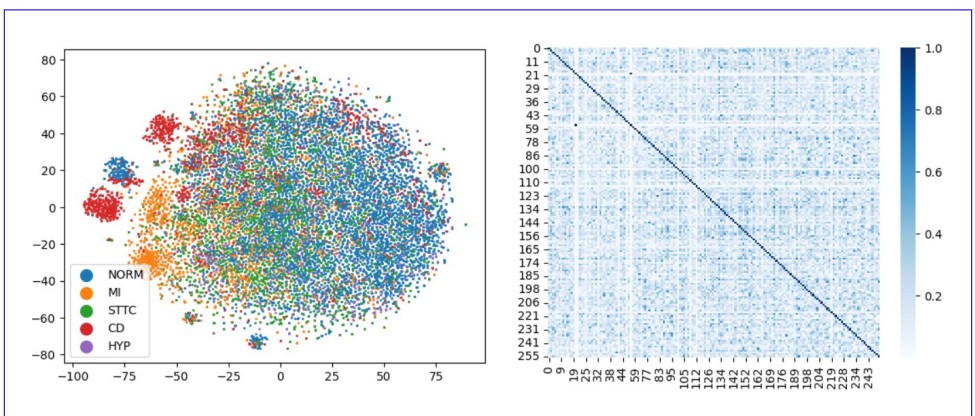

Figure 4: The *TA-PCLR* features using self supervised pretraining: Left) The t-SNE plots for the PTB-XL super classes. Right) Correlations between features.

## D  SUPERVISED TRAINING HYPER-PARAMETER

Details of the supervised training hyper-parameters are presented in Table 9

Table 8: Additional results for the PTB-XL Super and sub classes classification for $TA\text{-}PCLR$.

| No. | label | precision | recall | F1 | accuracy | AUROC |
|---|---|---|---|---|---|---|
| | | | Linear Probe | | | |
| 1 | Super | $0.752\pm0.0027$ | $0.578\pm0.0035$ | $0.643\pm0.0030$ | $0.860\pm0.0006$ | $0.889\pm0.0003$ |
| 2 | Sub | $0.531\pm0.0193$ | $0.316\pm0.0067$ | $0.369\pm0.0063$ | $0.961\pm0.0002$ | $0.912\pm0.0015$ |
| 3 | MI | $0.783\pm0.0039$ | $0.811\pm0.0043$ | $0.794\pm0.0020$ | $0.837\pm0.0035$ | $0.895\pm0.0013$ |
| 4 | STTC | $0.761\pm0.0016$ | $0.808\pm0.0024$ | $0.777\pm0.0014$ | $0.823\pm0.0020$ | $0.885\pm0.0004$ |
| 5 | CD | $0.754\pm0.0070$ | $0.801\pm0.0046$ | $0.770\pm0.0057$ | $0.823\pm0.0073$ | $0.888\pm0.0018$ |
| 6 | HYP | $0.681\pm0.0031$ | $0.800\pm0.0044$ | $0.711\pm0.0037$ | $0.835\pm0.0054$ | $0.867\pm0.0016$ |
| 7 | NORM | $0.832\pm0.0014$ | $0.835\pm0.0014$ | $0.825\pm0.0022$ | $0.825\pm0.0022$ | $0.917\pm0.0002$ |
| | | | Fine tuning | | | |
| 8 | Super | $0.777\pm0.0062$ | $0.681\pm0.0080$ | $0.720\pm0.0041$ | $0.884\pm0.0010$ | $0.919\pm0.0007$ |
| 9 | Sub | $0.534\pm0.0218$ | $0.376\pm0.0218$ | $0.420\pm0.0082$ | $0.964\pm0.0005$ | $0.928\pm0.0029$ |

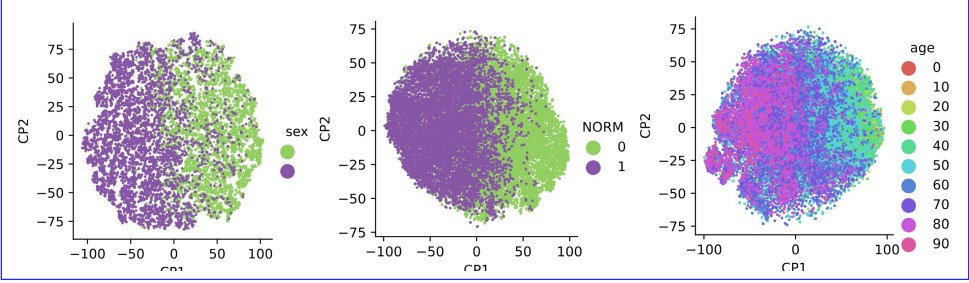

Figure 5: The $TA\text{-}PCLR$ features using self supervised pretraining: Left) The t-SNE plots for the PTB-XL super classes. Right) Correlations between features.

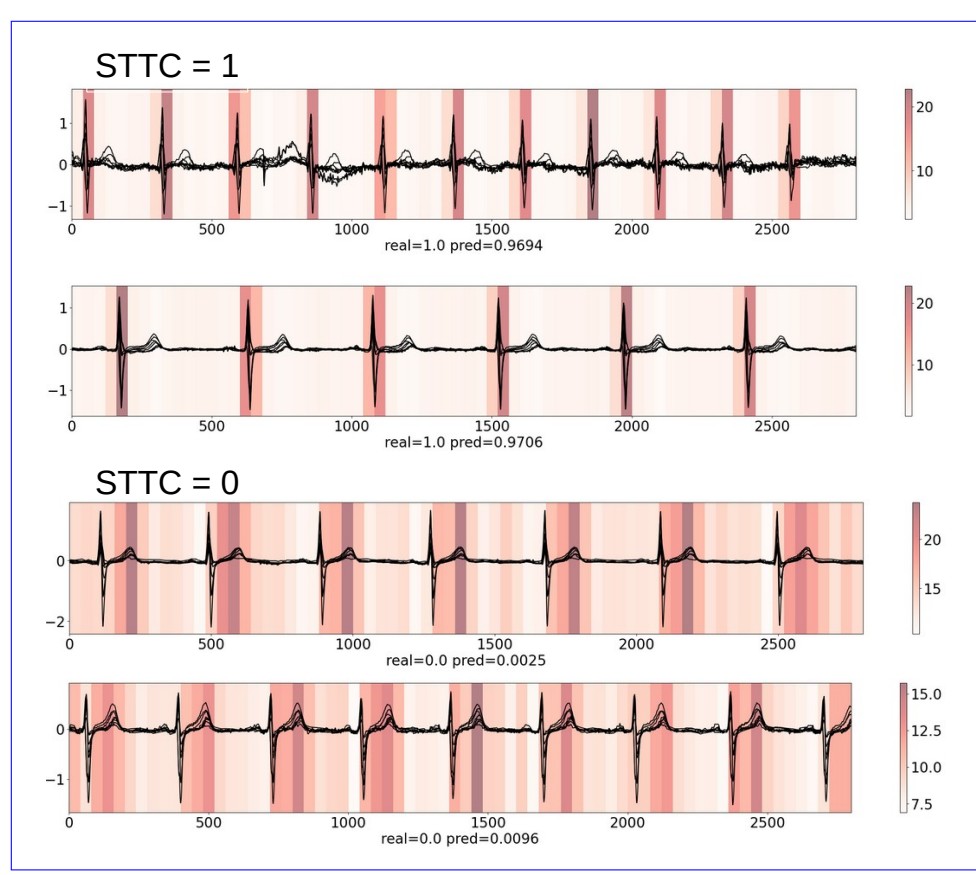

Figure 6: The GradCam is used to highlight the regions contributing to the prediction of a label. The ECGs are superimposed on the gradient maps with darker colors indicating more important regions. It can be observed that the model gives high importance to the ST segment () for normal samples while failing to do so in the abnormal samples, thus probably using the absence of the pattern as the indication of the diagnosis.

Table 9: Training hyperparameters

| No. | task | Parameter | Value |
|---|---|---|---|
| 1 | All | optimizer | Adam with lr schedule: reduce-on-plateau and early-stopping |
| 2 | ResNet (all) | learning rate | 0.0005 |
| | | Section 4.1 and 4.2 | |
| 2 3 | Age regression | learning rate prediction head | 0.0001 MLP hidden = [256, 128] |
| 4 5 | Sex classification | learning rate prediction head | 0.0001 MLP hidden = [256, 256] |
| 6 7 | Mortality (5y) classification | learning rate prediction head | 0.0001 MLP hidden = [256, 256] |
| | | Section 4.2 Table 3 | |
| 8 9 | Age regression | learning rate prediction head | 0.005 single neuron |
| 10 11 | Sex classification | learning rate prediction head | 0.005 single neuron |
| 12 13 | Super classes classification | learning rate prediction head | 0.0001 single layer with neurons equal to the number of outputs |
| 14 15 | Sub classes classification | learning rate prediction head | 0.0001 single layer with neurons equal to the number of outputs |
| | | Section 4.3 Table 4 | |
| 16 17 | Super classes regression | learning rate prediction head | 0.005 single neuron |

