# OpenReview forum: "Electrocardiogram Foundation Model Using Temporally Augmented Patient Contrastive Learning"
_ICLR.cc/2025/Conference — Submitted to ICLR 2025_

### Official Review · Reviewer_T2x4 · 2024-11-03

**Soundness:** 2
**Presentation:** 3
**Contribution:** 2
**Rating:** 1
**Confidence:** 5

**Summary:**

This paper proposes a novel Electrocardiogram Foundation Model using Temporally Augmented Patient Contrastive Learning (TA-PCLR), focusing on building a large-scale FM using ECG data. The TA-PCLR model combines patient-based contrastive learning with temporal augmentations, and it demonstrates superior results over traditional fully-supervised models.

**Strengths:**

Compared to other ECG FM studies, a key strength of this paper is the utilization of large-scale data. By handling over six million ECGs, the model incorporates diverse demographic characteristics, creating favorable conditions for improved model generalization.

**Weaknesses:**

The paper appears to lack analytical depth. Adding an exploratory data analysis (EDA) would provide a more comprehensive understanding of data distribution and representation, which could, in turn, enhance the reliability of the results. Additionally, assessing the representation distribution across different data types could further substantiate the FM’s robustness. For example, first, using methods like PCA or t-SNE, visualize the embeddings generated by the model in 2D or 3D to examine whether different data types fall within similar distributions or are distinctly separated. This analysis helps assess if the model represents various data types consistently. Second, analyze whether different data types are well-mixed or independently clustered to understand if the model extracts consistent features across types or demonstrates bias toward certain types. Third, examine specific features the model prioritizes across data types (e.g., certain words in text, patterns in images, frequency ranges in signals) to determine if the model consistently learns representations across data types. Lastly, feature correlation analysis assesses the relationships between key features learned by the model, which is especially insightful for multimodal data to understand how the model connects specific features across types.

The model primarily relies on temporal transformations such as zero-masking and cropping, with limited exploration of other data augmentation techniques. Moreover, the training approach employs conventional contrastive learning without attempting novel methods. The choice to rely solely on masking lacks sufficient justification, making it challenging to fully accept this design decision. Given the reliance on a limited set of augmentation techniques (https://dl.acm.org/doi/10.1145/3511808.3557591, https://arxiv.org/abs/2204.04360), it is difficult to assess their true effectiveness. To strengthen the study and provide clearer evidence of the augmentation strategy’s impact, it would be beneficial to incorporate and compare additional methods discussed in previous research, as outlined in this paper and this paper. Such a comparative approach would provide deeper insights and enhance the robustness and reliability of the findings regarding the model’s performance.

**Questions:**

To substantiate the proposed approach, it is essential to compare and analyze its performance against various existing learning approaches. Currently, the model relies on contrastive learning; however, evaluating its performance relative to other frameworks, such as Masked Autoencoder (MAE), could provide valuable insights. Specifically, understanding how TA-PCLR performs in comparison to MAE or other non-contrastive self-supervised learning methods would offer a clearer picture of its strengths and limitations in feature extraction and representation learning.

From a technical perspective, additional experiments are proposed to further validate the approach. Comparing the model’s performance against currently available models, such as those discussed in (https://arxiv.org/abs/2408.05178), would provide a solid benchmark. Additionally, applying alternative contrastive learning techniques, such as MoCo, could reveal how different contrastive frameworks impact model performance. It would also be insightful to experiment with exclusively using MAE as a non-contrastive method to better understand the unique contributions of the contrastive approach in this context.

By conducting these comparative experiments with a range of learning frameworks, this study could offer a more comprehensive evaluation, showcasing both the effectiveness of TA-PCLR and the relative benefits of its contrastive learning strategy. Such an analysis would greatly enhance the reliability and depth of the findings regarding the model’s capabilities in feature extraction and representation learning.
A comparative analysis would allow us to assess whether TA-PCLR’s temporal augmentations and patient-based contrastive learning truly offer a competitive advantage over alternative methods. Such comparisons could reveal potential improvements in the model’s predictive accuracy, robustness, and generalization across different ECG data subsets.

---

> ### Author Response · Authors · 2024-11-25
> **Response 1 to Reviewer T2x4**
>
> We highly appreciate your insightful comments and value your interest in indicating weaknesses in our work. You highlighted important research questions that we carefully tried to address. We regret that our work was not satisfactory for your approval previously but we have greatly revised the paper organization and presentation that we would be thankful for you to go through. We would be grateful for your time to read our responses and view the revised manuscript.
>
> We have restructured Table 2 and replaced the previous Table 3 with a graph. We have tried to rephrase where our previous explanation could be confusing or misleading and provided additional results in Appendix A, B, and C that might address most of your concerns and we are hopeful will result in revising your previous assessment. We will now go through each comment and present our response and the corresponding changes in the new version of our paper.
>
> **Reviewer comment:**
> The paper appears to lack analytical depth. Adding an exploratory data analysis (EDA) would provide a more comprehensive understanding of data distribution and representation, which could, in turn, enhance the reliability of the results.
>
> **Response:**
> We have added some demographics of the datasets in Appendix A Table 6, and we also provide references in Section 3.1 where further details can be obtained.
>
> **Reviewer comment:**
> Additionally, assessing the representation distribution across different data types could further substantiate the FM’s robustness. For example, first, using methods like PCA or t-SNE, visualize the embeddings generated by the model in 2D or 3D to examine whether different data types fall within similar distributions or are distinctly separated. This analysis helps assess if the model represents various data types consistently.
> Second, analyze whether different data types are well-mixed or independently clustered to understand if the model extracts consistent features across types or demonstrates bias toward certain types.
> Third, examine specific features the model prioritizes across data types (e.g., certain words in text, patterns in images, frequency ranges in signals) to determine if the model consistently learns representations across data types.
> Lastly, feature correlation analysis assesses the relationships between key features learned by the model, which is especially insightful for multimodal data to understand how the model connects specific features across types.
>
> **Response:**
> We have added some interpretability analysis in Appendix B for the PTB-XL dataset. We would like to highlight that our model is self-supervised so the learned representations are generic and thus suitable to a foundation model. The tasks are multi-label and composite so the same ECG can manifest different conditions and the same label encompasses several modes. Figure 4 shows how the principle components of the embedding encode different superclasses. It can be appreciated that the embedding space places similar ECGs in nearby locations. Some of the composite classes can be observed to form several clusters. The ECG encoding may not be limited to only cardiac diseases but all the underlying structure distribution. Figure plots tsne components for only those features that highly correlate to a given label sex, age, and normal/abnormal. The different classes fall on opposite sides of PC1 while a regression target like age shows a gradient. We address your third comment by Figure 6 using Gradcam to show how the model represents different classes. STTC class is related to the ST segment of an ECG so we look at some positive and negative samples. It was interesting to note that the model is classifying by giving more importance to the ST segment for negative classes (normal). It is also more intuitive that instead of recognizing several abnormal modes it finds it more efficient to recognize the normal ST segment. We have added a feature correlation map in Figure 5 to signify how the learned features relate to each other.
> **Reviewer comment:**
> The model primarily relies on temporal transformations such as zero-masking and cropping, with limited exploration of other data augmentation techniques.
>
> **Response:**
> The selection of temporal augmentation is based on the fact that the signal in this case is important biological data with clinical implications for the scale, frequency, and underlying patterns so augmentations like scaling, warping, rotating, etc may result in features that will be agnostic to important clinical information for a potential downstream task thus the model will loose generalization that is an important aspect of FM. We experimented with some variations of the masking and also tried a novel augmentation using raw/filtered ECGs. We now include these results in Table 2.

---

> ### Author Response · Authors · 2024-11-25
> **Response 2 to Reviewer T2x4**
>
> **Reviewer comment:**
> Moreover, the training approach employs conventional contrastive learning without attempting novel methods.
>
> **Response:**
> The particular loss function employed resulted in robust performance and far surpassing a much larger supervised and pretraining approach. In the scope of current work, we limit ourselves to our best-performing representation learning approach although we did experiment with variations of the loss function, but the performance was not comparable. We have previously published work employing triplet loss and VAE.
>
> **Reviewer comment:**
> The choice to rely solely on masking lacks sufficient justification, making it challenging to fully accept this design decision. Given the reliance on a limited set of augmentation techniques (https://dl.acm.org/doi/10.1145/3511808.3557591, https://arxiv.org/abs/2204.04360), it is difficult to assess their true effectiveness. To strengthen the study and provide clearer evidence of the augmentation strategy’s impact, it would be beneficial to incorporate and compare additional methods discussed in previous research, as outlined in this paper and this paper. Such a comparative approach would provide deeper insights and enhance the robustness and reliability of the findings regarding the model’s performance.
>
> **Response:**
> The ECG signal is a snapshot of a biological process where any augmentation cannot be safely incorporated as the training process discards features encoding the augmentation dimension. The selection of the particular augmentations is based on the fact that such random masking and cropping do not change the signal characteristics and improve the generalization of the approach which is an important consideration for a foundation model. The two publications that you suggest (https://dl.acm.org/doi/10.1145/3511808.3557591, https://arxiv.org/abs/2204.04360) also strengthen our conviction that the use of augmentation cannot be applied without an understanding of the tasks. These research endeavors are focussed on data augmentation for a particular supervised task and while some augmentations might improve the performance on one task, it may have adverse effects on others. https://arxiv.org/abs/2204.04360 in particular notes that zero-masking being “label preserving”, provides a more robust performance across a range of labels. We also experimented with random masking across leads, random lead masking, and using unfiltered signals as augmentations but the performance did not change so we retained the simpler configuration. We have revised Section 3.3 which defends and explains our choice of augmentations. We have also added the suggested references with the following explanation:
>
> *Section 2:
> “Past work has shown that while some augmentations might improve the performance on one task, they may have adverse effects on others (Lee et al., 2022; Raghu et al., 2022).”
> *
>
> **Reviewer comment:**
> To substantiate the proposed approach, it is essential to compare and analyze its performance against various existing learning approaches. .....Comparing the model’s performance against currently available models, such as those discussed in (https://arxiv.org/abs/2408.05178), would provide a solid benchmark.
>
> **Response:**
> In the main paper, we only limited comparisons to where the original publications report a metric. A lot of performance gain can be obtained by proper optimisation so we think it is unfair to report our version that may not be as well optimised.  Table 5 includes  Song et al. (2024) (https://arxiv.org/abs/2407.07110) which is a foundation model trained by a hybrid MAE+ contrastive learning. We have also added some performance comparisons with a range of other pretraining methodologies in Appendix B, Table 7 concluding MAE-based approaches like STMEM (https://iclr.cc/media/iclr-2024/Slides/18470.pdf).
>
> **Reviewer comment:**
> Additionally, applying alternative contrastive learning techniques, such as MoCo, could reveal how different contrastive frameworks... model’s predictive accuracy, robustness, and generalization across different ECG data subsets.
>
> **Response:**
> We have tried to address all of your concerns but adding Appendix B Table 7 and restructuring Table 2 as an ablation study for augmentations. We have also added interpretability research and we hope that you can appreciate the performance edge that the proposed method has and how we have integrated clinical understanding into our model design. While in future work we can further explore if the loss function can be further optimized.

---

### Official Review · Reviewer_poSx · 2024-11-03

**Soundness:** 3
**Presentation:** 3
**Contribution:** 3
**Rating:** 6
**Confidence:** 5

**Summary:**

Summary
The study introduces TA-PCLR (Temporally Augmented Patient Contrastive Learning for ECG Representations), a self-supervised learning approach for ECG data. This method combines temporal augmentations and patient-based contrastive learning to improve representation learning in ECGs. Key highlights include:

1. Novelty of TA-PCLR: It introduces temporal augmentations, such as zero-masking and random cropping, to enrich representations while preserving clinical relevance.
2. Performance Improvements: TA-PCLR outperforms previous methods in tasks such as predicting age, sex, and five-year mortality from ECG data, especially on small and medium-sized datasets where labeled data is scarce.
3. Impact of Dataset Demographics: The study explores how population diversity in training datasets influences model performance, showing that a mixed, diseased cohort provides richer ECG feature diversity and enhances model performance.

In indipendent validation with the PTB-XL dataset, TA-PCLR surpasses other models in diagnostic tasks, establishing itself as a robust foundation model for ECG interpretation. Future directions include refining data augmentations and enhancing interpretability in ECG feature learning.

**Strengths:**

- Effective Representation Learning: TA-PCLR leverages temporal augmentations with patient-based contrastive learning to enhance feature extraction. This approach enables more nuanced and clinically relevant ECG representations, which are critical for downstream diagnostic tasks, even in data-scarce environments.
- Broad Applicability and Generalization: Training on a diverse dataset of over 6 million ECGs from multiple global cohorts makes TA-PCLR highly generalizable. This diversity allows the model to perform well across varied patient demographics and health conditions, ensuring its robustness across different healthcare settings.
- Superior Performance on Limited Data: The model’s ability to perform better than fully supervised methods on small to medium-sized labeled datasets makes it especially valuable in clinical contexts where labeled data is often scarce. This makes TA-PCLR a powerful tool for institutions with limited data resources.

**Weaknesses:**

The important contribution of this research is that it highlights the importance of geographical and racial diversity in training data when creating machine learning models for electrocardiograms, but on the other hand, the results of this research have only been verified using the PTB-XL dataset.
The scope of this study may be narrow than the general interest of ICLR main conference.

**Questions:**

Do the authors have any specific proposals for the above weaknesses? How can we build the necessary data sets to verify geographical and racial diversity?

---

> ### Author Response · Authors · 2024-11-25
> **Response to Reviewer poSx**
>
> We greatly appreciate your encouraging reviews and discussion about the weaknesses. We believe that the work is pertinent to the interest of the ICLR due to the discussion and insights for representation learning in medical applications. The design of our approach is inspired from a clinical perspective and we have further improved the overall organization and rephrased where working could be confusing or misleading. We have also updated Table 2 with a comparison of several variations of the temporal masking that we experimented with. We also replace Table 3 with Figure 2. Additional comparisons with other approaches are incorporated in Appendix B Table 7 with detailed metrics in Table 8. Some preliminary work on interpretability is also incorporated in Appendix D Figures 4 to 6 that may be of great interest to the ICLR conference. We will be highly grateful for your time and interest in going through the revised paper and reconsidering your previous score.
>
> **Reviewer comment:**
> The important contribution of this research is that it highlights the importance of geographical and racial diversity in training data when creating machine learning models for electrocardiograms, but on the other hand, the results of this research have only been verified using the PTB-XL dataset. The scope of this study may be narrow than the general interest of ICLR main conference.
> Do the authors have any specific proposals for the above weaknesses? How can we build the necessary data sets to verify geographical and racial diversity?
>
> **Response:**
> The contributions of our work are multi-faceted, based on not only the exploration of the importance of geographical and racial diversity in both the pretraining and labeled datasets but also the foundation model that is highly essential for the particular domain where labeled data is highly expensive. The PTB-XL dataset is mainly employed for comparison to past literature but the specific aspect of the diverse dataset is explored in Section 4.2 where we have designed a series of experiments to investigate. The experiments show that the performance is impacted by the dataset diversity but we prove that our multi-centered dataset makes our model robust to the diverse distribution, consistently performing best for all labeled datasets.

---

> > ### Comment · Reviewer_poSx · 2024-11-27
> >
> > After reading all the reviewers' comments and the author's response, I could clearly understand that the author places importance on a research approach that emphasizes clinical application. This is an advantage from the perspective of those who emphasize application, but it is a weakness from the perspective of those who emphasize theory, but it is difficult to achieve a perfect balance within a single paper. I will maintain the current score.

---

### Official Review · Reviewer_7RW1 · 2024-11-04

**Soundness:** 3
**Presentation:** 2
**Contribution:** 1
**Rating:** 1
**Confidence:** 5

**Summary:**

This paper presents an ECG-based foundation model using a large-scale dataset of six million ECGs from different institutions. The authors propose to use zero masking and random cropping as augmentation strategies.

**Strengths:**

1. The main strength of the paper is the scale of the dataset that they have access to, which combines data from multiple cohorts.
2. The aim of building an ECG foundation model is interesting and important.
3. The authors evaluate their proposed framework for multiple downstream prediction tasks.

**Weaknesses:**

1. The proposed work lacks novelty. The framework does not possess originality expected by ICLR and is very similar to existing frameworks. In fact, it is unclear to me what the novelty is. The methods section presents the vanilla InfoNCE loss and then discusses the augmentations, preprocessing, architecture and training scheme.
2. Comparison to existing SOTA frameworks is very limited. The authors do mention relevant papers but only compare to a few in the results section. There are many notable works in this area that are worth comparing to.
3. Did the authors conduct hyperparameter tuning for their proposed framework and baselines? What was the approach? I appreciate that the final values are listed in the appendix however this is not sufficient.
4. The authors did not conduct any statistical significance testing nor did they provide confidence intervals to understand whether the performance improvements are actually worthwhile.
5. Numbers should be presented with three significant figures.
6. There are empty rows in Tables 4 and 6, why is that?

**Questions:**

Please see above.

---

> ### Author Response · Authors · 2024-11-25
> **Response to Reviewer 7RW1**
>
> We highly appreciate your time and careful review. Your suggestions allowed us to learn the weaknesses of our research and helped to improve the quality of the work. We regret that you found the work to be lacking in many respects and we will try our best to address your concerns. We have substantially revised our work, with additional results and rephrasing where the wording could be confusing or misleading. Your main concern is novelty so we wish to explain that the research would be of great interest to representation learning and specifically to that involving electrocardiogram data. We demonstrate remarkable performance improvement to prior research since the use of augmentation was inspired from a clinical perspective. The main paper limits the comparison to only research which reports performance for PTB-XL dataset, removing bias due to lack of optimization. We add Table 7 in Appendix B providing a wider comparison with a range of past works and additional metrics and statistics in Table 8. Another main aspect is the use of a large multi-center cohort that allowed us to make further insights about the impact of underlying data distribution on the pre-training and subsequent supervised tasks that can be of great value for the particular field. We also incorporate some interpretability study in Appendix C with further discussion about how the model encodes ECG features. We hope that you will consider revising your previous score.
>
> **Reviewer Comment:**
> The proposed work lacks novelty. The framework does not possess originality expected by ICLR and is very similar to existing frameworks. In fact, it is unclear to me what the novelty is. The methods section presents the vanilla InfoNCE loss and then discusses the augmentations, preprocessing, architecture and training scheme.
>
> **Response:**
> Our approach uniquely combines components from existing literature from the perspective of clinical implications and employs a large multi-center pretraining cohort to present a foundation model that is highly essential to the particular application domain where labeled data is scarce. We outperform highly optimised ensemble of supervised networks, as well as more complex pretraining approaches thus proving the efficiency and strength of the approach. That poses our foundation model as an interesting contribution to the domain of ECG interpretation where the performance is often limited by the small size of labeled datasets.
>
> **Reviewer Comment:**
> Comparison to existing SOTA frameworks is very limited. The authors do mention relevant papers but only compare to a few in the results section. There are many notable works in this area that are worth comparing to.
>
> **Response:**
> We limited the performance comparison in the main paper to research that reports performance for the PTB-XL labels, as the performance greatly depends on the optimisation of hyperparameters for both the pretraining and downstream tasks. We believe that it is not fair to compare with our implementation of other approaches that may not be as thoroughly optimised. To provide more comparability we have added Table 7 in Appendix B comparing the performance with a range of other methodologies. The proposed approach demonstrates a much higher performance as compared to other unsupervised pretraining approaches.
>
> **Reviewer Comment**:
> Did the authors conduct hyperparameter tuning for their proposed framework and baselines? What was the approach? I appreciate that the final values are listed in the appendix however this is not sufficient.
>
> **Response:**
> We have incorporated further details in all relevant sections. A hyper-parameter optimisation was not performed for our work and mainly learning rate was manually optimised within a range of [0.1, 0.00001] for supervised tasks while the backbone architecture is similar to past works (https://journals.plos.org/ploscompbiol/article?id=10.1371/journal.pcbi.1009862) for a fair comparison.
>
> **Reviewer Comment:**
> The authors did not conduct any statistical significance testing nor did they provide confidence intervals to understand whether the performance improvements are actually worthwhile.
> Numbers should be presented with three significant figures.
> There are empty rows in Tables 4 and 6, why is that?
>
> **Response:**
> For all experiments conducted, the performance is reported as a mean of ten independent runs similar to the available literature. We provide three significant figures for our experiments and the available values from past work. We have added additional results in Appendix B Table 8 that provide both mean and 95% confidence intervals for multiple metrics, for all results reported in Section 4.3.

---

> > ### Comment · Reviewer_7RW1 · 2024-11-28
> >
> > Thank you for addressing my questions and concerns. Unfortunately I will not change my score considering that the authors did not conduct hyperparameter tuning, limited comparisons to SOTA work, and due to lack of novelty.

---

> > > ### Author Response · Authors · 2024-11-28
> > > **Response**
> > >
> > > We highly respect your judgment and hope that you had time to go through the updated manuscript where we tried to incorporate all your suggestions. We would be highly grateful if you could read the final manuscript (changes in blue) and the following discussion.
> > >
> > > **Hyperparameter optimization:**
> > > I apologize for misunderstanding "Hyperparameter optimization" as submitting an automatic search that our resources did not allow. I wish to clarify, that we experimented with several hyperparameter settings through manual search due to long training times, similar to many past works. We share the best-performing network hyperparameters as the paper limits did not allow us to go through all settings. For the downstream linear probe, the learning rate was the only hyperparameter to explore that we optimised within a range of [0.1, 0.00001]. The architecture backbone has been previously optimized for ECG classification in past works. We optimised the lr for pretraining, feature size, window size, and MLP architecture for results in section 4.1.
> > >
> > > **Comparison to SOTA approaches**. We have compared our work to a wider range of methodologies in new **Table 7** (we explain that comparison in Table 3 and 4 is limited to those publications that present a result for the particular target). Our results outperform all previous works, whether supervised or using pretraining, for much more complex models and approaches, including ensembles. That really attests to the fact that our contrastive strategy takes into account a clinical understanding.
> > >
> > > **novelty**. The particular configuration that we use has not been used before. We design the augmentation by considering the clinical implications of the transformations and the success of the approach is attested by the performance. Similarly, the experiments for data diversity would also provide a new perspective that is much needed in the field. The ICLR scope also includes "applications to physical sciences (physics, chemistry, biology, etc.)" so we hope that our results and insights will be pertinent to the ICLR scope.

---

### Official Review · Reviewer_hUMx · 2024-11-04

**Soundness:** 2
**Presentation:** 2
**Contribution:** 1
**Rating:** 3
**Confidence:** 4

**Summary:**

The study aimed to develop a contrastive learning based foundation model for ECG data to predict patients Age, Sex, 5-year mortality and various cardio abnormalities. The authors claimed that they have proposed a novel approach by introducing temporal augmentation to the previously proposed contrastive learning algorithm PCLR. Experiments with the major ECG datasets that are publicly available were conducted to benchmark the performance of the proposed method to some of the existing methods.

**Strengths:**

Development of foundation models for medical data and signals is obviously an important topic worth studying. Additionally, it is also true that the cost for engaging physicians to provide high quality labels for medical data is indeed much higher than many other domains. I can see how this can motivate the use of contrastive learning on medical data analysis.

**Weaknesses:**

1/ Lack of originality and novelty

The proposed method is built upon the previously published contrastive learning backbone, PCLR (by Diamant, Nathaniel, et al in 2022). The only enhancement was the introduction of some temporal augmentation of the ECG signals. These temporal augmentations included zero-marking and random clipping. These augmentations were nothing new too. Also, none of these augmentations were specifically inspired by any in-depth understanding of the ECG signals and its unique characteristics.

A commonly used loss function was used as well.

I failed to the see the technical originality and contributions here.

Can the authors highlight and justify their technical novelty and contributions in this work further?

2/ Possible inflating some information
The authors claimed they trained their foundation model over 6 Mil ECGs in the introduction. After reading into the details then I learned that the authors actually counted a 12-lead ECG measurements of a single patient 12 ECG signals. This is a bit unusual. Can't help but to guess whether the team might present the dataset size in an inflated manner?

3/ Poorly designed experiments

The experimental design was so fragmented and confusing. There lacked of a systematic framework and experimental design to fairly benchmark the proposed method over the SOA methods over various tasks and datasets.

In Table 2, the authors compared PCLR with the proposed TA-PCLR. But PCLR were only pre-trained over the MGH and BIDMC datasets but not the combined BCSV dataset. The BCSV combined dataset is the largest. Why didn't a head-to-head comparison was done for PCLR and TA-PCLR for the BCSV dataset?

In Table 3, the authors then compared the proposed TA-PCLR against ResNet. Why ResNet, quite an old model, was picked as a baseline model to be compared with here? Why didn't the performance of PCLR included here?

In Table Table 4, the authors then compared TA-PCLR with the ensemble model developed by Strodthoff et al. in 2021 and the model in Bickmann et al. (2024) for the classification for both the super classes and sub classes for abnormalities. However, many readings for the result table were missing. What happened?

In Table 5, similarly, many of the result readings were missing. What happened?

It felt like the team didn't manage to finish the study in time and yet just submit the manuscript before the deadline?

**Questions:**

Please refer to the above section for my questions.

---

> ### Author Response · Authors · 2024-11-25
> **Response 1 to Reviewer hUMx**
>
> We are highly grateful for your insightful comments and your time and interest. Your indication of weaknesses greatly helped to enhance the quality of our work and especially the last comment was very amusing. We present a unique combination of existing components that is inspired by a clinical understanding of ECGs. The strength of our work is the high level of performance, against orders of magnitude more complex networks. Our unique dataset allowed us to explore the effect of the dataset diversity and its impact on downstream tasks that can be highly relevant to the domain, as label-based comparisons are common in literature. We demonstrate the superiority of our mult-center dataset for improved robustness and generalization for a range of diverse data distributions. We have rephrased and reorganized the manuscript to more effectively describe the approach. Table 2 is restructured to remove confusion and added results for different variations of the temporal augmentations and a novel augmentation of contrasting by raw vs. filtered that we had experimented with. The previous Table 3 is converted to Figure 2 which compares the proposed approach to a fully supervised network with a similar backbone. A new section was added as Appendix B that provides a wider comparison to the state of the art implementations where our approach outperforms much complex pretraining and even a zero-shot approach trained incorporating ECG reports in Table 7 (adding some notion of supervision). Appendix C explore interpretibility of the learned representations and provides interesting insights. We hope that you will go through our responses and revised manuscript and would reconsider your previous score.
> **Reviewer comment:**
> 1/ Lack of originality and novelty
>
> The proposed method is built upon the previously published contrastive learning backbone, PCLR (by Diamant, Nathaniel, et al in 2022). The only enhancement was the introduction of some temporal augmentation of the ECG signals. These temporal augmentations included zero-marking and random clipping. These augmentations were nothing new too. Also, none of these augmentations were specifically inspired by any in-depth understanding of the ECG signals and its unique characteristics.
>
> A commonly used loss function was used as well.
>
> I failed to the see the technical originality and contributions here.
>
> Can the authors highlight and justify their technical novelty and contributions in this work further?
>
> **Response:**
> We have combined existing building blocks in a novel configuration that is inspired by an understanding of ECGs. Other works take generic time-series augmentations and apply that to ECG which is a biological signal. The scale, axis and frequency are important information and thus scaling, rotating, warping, and permutation can alter the underlying cardiac characteristics. Apart from different variation of the masking we also experiment with a novel augmentation of raw vs. filtered ECGs. We have focused on augmentations that do not alter any physical characteristics and trained the model on a large multi-center dataset added to Table 2. We have updated the description in Section 3.3 to more clearly explain the motive for the design decisions.
>
> The remarkable performance of the approach attests to the relevance of our hypothesis and will be very interesting to the representation learning community. The pretraining not only outperforms more complex pretraining but also a highly optimised ensemble of supervised networks with a much larger number of parameters. Our foundation model is an important contribution to the domain of ECG interpretation where the performance is often limited by the small size of labeled datasets. We also investigate the importance of geographical and racial diversity in training data for both the pretraining and supervised tasks, which will be very interesting for the medical domain where the results are often based on labels evaluated for diverse datasets.
>
> **Reviewer comment:**
> 2/ Possible inflating some information The authors claimed they trained their foundation model over 6 Mil ECGs in the introduction. After reading into the details then I learned that the authors actually counted a 12-lead ECG measurements of a single patient 12 ECG signals. This is a bit unusual. Can't help but to guess whether the team might present the dataset size in an inflated manner?
>
> **Response:**
> The training is performed for more than six million individual ECGs. The 12 leads are not counted as 12 but as a single ECG similar to past literature. We only utilize eight leads as the rest of the leads are only combinations of other leads. We do not know how the paper can be interpreted to relate the data size to leads but we further clarify by adding the following in Section 3.1:
>
> “We denote the combined pretraining cohort as BCSV consisting of more than six million individual ECGs, with each ECG comprising eight leads.”

---

> > ### Author Response · Authors · 2024-11-25
> > **Response 2 to Reviewer hUMx**
> >
> > **Reviewer comment:**
> >
> > 3/ Poorly designed experiments
> >
> > The experimental design was so fragmented and confusing. There lacked of a systematic framework and experimental design to fairly benchmark the proposed method over the SOA methods over various tasks and datasets.
> >
> > **Response:**
> > The experimentation can be divided into a proof of concept in Section 4.1 that performs some ablation study using the supervised tasks labels from the pre-training dataset. Section 4.2 provides additional investigation regarding the impact of racial and health diversity on both the pre-training cohort and labelled dataset that provides valuable insights for future work. Section 4.3 is limited to comparison to SOTA techniques that report performance for PTB-XL since it will not be fair to compare our implementationof other approaches that may be not be sufficiently optimised but we incorporated an additional Appendix B that compares our approach to a wider range of pretraining applications including approaches that use auxiliary information like ECG reports to provide zero-shot inference. We believe that integrating such information adds bias that can hurt the generalization that is important for a foundation model. The proposed approach outperforms all techniques.
> >
> > **Reviewer comment:**
> > In Table 2, the authors compared PCLR with the proposed TA-PCLR. But PCLR were only pre-trained over the MGH and BIDMC datasets but not the combined BCSV dataset. The BCSV combined dataset is the largest. Why didn't a head-to-head comparison was done for PCLR and TA-PCLR for the BCSV dataset?
> >
> > **Response:**
> > We modified Table 2 as the table mainly represented an ablation study of the approach and the comparison with the pretrained PCLR was only confusing the reader. We  add results for other variations of temporal augmentations that we had explored. The table intends to show how each component improves the performance so for the same configuration the TA-PCLR performs better than PCLR. Finally, we take the best configuration and train it with the larger dataset. We also restructure and update the explanation in Section 4.1 accordingly. Due to long training time the model development was only undertaken with BIDMC and only the final optimum model was pretrained with the larger dataset.
> >
> > **Reviewer comment:**
> > In Table 3, the authors then compared the proposed TA-PCLR against ResNet. Why ResNet, quite an old model, was picked as a baseline model to be compared with here? Why didn't the performance of PCLR included here?
> >
> > **Response:**
> > We replaced Table 3 with Figure 2 and present the comparison through plots. We compare the performance of TA-PCLR with supervised training for a similar model as the backbone architecture. The architecture was based on Resnet for comparison to PCLR but as the model outperforms supervised training and pretraining for models with orders of magnitude more parameters we consider the architecture to be adequate for expressing the ECG features. We prove in Table 2 the performance of each design component that also involves patient-based augmentation (PCLR). The datasize comparison to supervised approach shows how the performance is especially relevant for low data size.
> >
> > **Reviewer comment:**
> > In Table Table 4, the authors then compared TA-PCLR with the ensemble model developed by Strodthoff et al. in 2021 and the model in Bickmann et al. (2024) for the classification for both the super classes and sub classes for abnormalities. However, many readings for the result table were missing. What happened?
> >
> > In Table 5, similarly, many of the result readings were missing. What happened?
> >
> > **Response:**
> > Table 4 and 5 includes the performance comparison for the PTB-XL classification only when reported in the original publication. That is the reason for the missing values in the tables. We have added ‘-’ to indicate these missing values and also tried to improve the explanation.
> >
> > **Reviewer comment:**
> > It felt like the team didn't manage to finish the study in time and yet just submit the manuscript before the deadline?
> >
> > **Response:**
> > We hope that our explanation of how the different experiments are performed will satisfy the reviewer to the completion of the research endeavor.

---

### Official Review · Reviewer_pv5E · 2024-11-04

**Soundness:** 2
**Presentation:** 2
**Contribution:** 2
**Rating:** 3
**Confidence:** 4

**Summary:**

This paper introduces Temporally Augmented Patient Contrastive Learning of Representations (TA-PCLR), a novel approach that incorporates temporal augmentations into a patient contrastive self-supervised foundation model.

**Strengths:**

1. This method adds temporally augmented patient contrastive learning that incorporates augmentations along the temporal axis.
2. This method constructed a new multi-center dataset for training with over six million ECGs with four cohorts from three different continents.

**Weaknesses:**

1. The organization of this paper needs improvement. The color of Fig2 is too dark. There are some empty results in table 4 and table 5, which should add a line '-' and explain why.
2. In table2, the pertaining cohorts in PCLR and TA-PCLR are different. One is MGH, and another is BCSV. It can't get any conclusion on the effectiveness of TA-PCLR because of using different cohorts.
3. The novelty of this method is concerned. Basically, this method just adds a general contrastive learning strategy to learn EEGs between patients.

**Questions:**

When showing performance comparison between Resnet and TA-PCLR, are there any latest SOTA methods like attention or transformer-based models to compare?

---

> ### Author Response · Authors · 2024-11-25
> **Response to Reviewer pv5E**
>
> We are highly grateful for your valuable comments and suggestions that helped us identify the weaknesses of our work and greatly enhance its quality. We have carefully studied all your comments and incorporated them into our revised manuscript. We have not only improved the paper organization and the overall text but we only performed additional experiments. We have reorganized Table 2 as an ablation study and added results from the exploration of different variations of our temporal augmentations together with novel raw vs filtered (clean) ECG contrasting. The Table 3 is converted to a graph format. Additional Appendices A, B, and C are added. Appendix A provides details about the demographics of the datasets used in the research. The comparisons for the external dataset PTB-XL in Section 4.3 is limited only to publications that report metrics for the classification tasks for the PTB-XL. Appendix B Table 7 provides a broader comparison with SOTA techniques that prove the superiority of our approach. Finally, Appendix C shares some insights from the interpretability investigation providing interesting observations. We would be grateful if you could go through our responses and our revision. We would be highly appreciative if you would reconsider your previous score.
>
> **Reviewer comment:**
> The organization of this paper needs improvement. The color of Fig2 is too dark.
>
> **Response:**
> We have updated the colors of Figure 2.
>
> **Reviewer comment:**
> There are some empty results in table 4 and table 5, which should add a line '-' and explain why.
>
> **Response:**
> Thanks for pointing out the omission. We have added dashes ‘-’ in the empty cells where the metric values are not available in the original publication. The performance greatly depends on the optimisation of hyperparameters for both the pretraining and downstream tasks so we believe that it is not fair to compare with our implementation of other approaches that may not be as thoroughly optimised. Although we have added a broader comparison in Table 7 (Appendix B).
>
> **Reviewer comment:**
> In table2, the pertaining cohorts in PCLR and TA-PCLR are different. One is MGH, and another is BCSV. It can't get any conclusion on the effectiveness of TA-PCLR because of using different cohorts.
>
> **Response:**
> We have restructured the table and removed the PCLR trained with MGH as it was confusing the readers. The initial intent was to show the impact of all components and PCLR with MGH was used as a baseline. The table now also presents the results for some variations of temporal augmentations that we experimented with.
>
> **Reviewer comment:**
> The novelty of this method is concerned. Basically, this method just adds a general contrastive learning strategy to learn EEGs between patients.
>
> **Response:**
>
> We use the existing augmentations in a novel combination inspired by clinical perspective and a large multi-center pretraining cohort to outperform other pretraining approaches. We were able to outperform a highly optimised ensemble of supervised networks with a much larger number of parameters attesting to the efficiency of the approach. That poses our foundation model as an interesting contribution to the domain of ECG interpretation where the performance is often limited by the small size of labeled datasets. Additionally, interesting research questions like the effect of dataset ethnic and health status diversity on the learned representations and the impact of the labeled data distribution on the performance are also discussed and interesting insights are shared.
>
> **Reviewer comment:**
> When showing performance comparison between Resnet and TA-PCLR, are there any latest SOTA methods like attention or transformer-based models to compare?
>
> **Response:**
> Section 4.1 is presented as a proof of concept thus a similar network is employed for both the supervised and our pretraining approach. The comparison to Resnet in Table 3 is now changed to a graphical format and Resnet is changed to supervised, to improve clarity. The comparison in Section 4.3 involves SOTA approaches including Strodthoff et al. (2021) (Table 3) a supervised approach involving an ensemble comprising larger Resnets, inception, LSTM, etc.  Song et al. (2024) compared in Table 4 is a transformer-based approach. We have added further performance comparison in Table 8 Appendix “Additional results for PTB-XL” which covers a range of pretraining methodologies with diverse architectures.

---

### Official Review · Reviewer_QAYi · 2024-11-08

**Soundness:** 3
**Presentation:** 2
**Contribution:** 2
**Rating:** 6
**Confidence:** 4

**Summary:**

The authors introduce a novel, ECG-specific self-supervised learning approach which incorporates temporal augmentations (random cropping) over patient contrastive methodology. This work is portrayed as a new foundation model for ECG interpretation, where experiments are run on a large, private pretraining dataset of six million unlabeled ECGs and involves a number of downstream tasks (age, sex, mortality, and diagnostic super-classes). They further examine how the pretraining dataset distribution impacts downstream performance by way of dataset-specific training and evaluation.

**Strengths:**

Their use of a multi-source pretraining dataset is certainly a strength. Their literature background gives a relatively strong, concise overview of existing pretraining approaches. Their methodology selection is, in large part, explained well and intuitively. Their data scaling was vital in demonstrating TA-PCLR's dominant performance over fully-supervised methods for smaller datasets. I have not personally seen the random cropping approach they introduced as their temporal augmentation, which speaks, in part, to its originality.

**Weaknesses:**

This paper is introducing a novel method more than it does a foundation model. Although they do pretrain on a large dataset, there is a lack of experimentation which only seems reasonable when releasing a foundation model. Namely, model scaling, considering how they adopt a relatively smaller architecture; They pride themselves on this point, however, this is unfounded due to a lack of confirmation that such a small deep neural network could actually make most effective use of the vast pretraining dataset. This would have been vital to confirm. The literature background refers to just a couple existing foundation models (and only in passing). There is no interpretability method. They also report just one metric per task, which greatly reduces study comparability. Considering these points, it seems lacking for a foundation model paper, even though it seems sufficient for simply introducing a novel pretraining method.

Considering this paper is introducing a new pretraining method, it may have benefited from an ablation study of its various pretraining components beyond a comparison to PCLR. Or, if the authors believe that this is all that's necessary as an ablation, then explaining this in more detail to better communicate their contribution.

Certain methodological matters are somewhat ambiguous. It was unclear whether hyperparameter tuning was performed. It is not made abundantly clear that the positive pairs are necessarily augmented instances of the same recording (and not from different recordings of the same patient). It is unclear whether there was any signal preprocessing (such as standardization), and if not, this may be worth stating.

Certain purported claims are left unsubstantiated, for example: "Human perception is limited by low visual accuracy, gaps in theoretical knowledge, and the complexity of the diverse, non-linear, interrelations."

Certain references may be misleading provided its context, for example, "Hybrid techniques combining contrastive learning and generative pretraining based on transformer architecture have been implemented to train foundation models for ECG feature extraction (Song et al., 2024; McKeen et al., 2024)" may imply that ECG-FM (McKeen et al., 2024) using generative pretraining, which is not true.

Certain definitions are questionable, for example, the statement that, "There are diverse contrastive learning approaches, differing in their definitions of the positive and negative instances and the loss computations" negates how certain contrastive methods do not apply here, such as using a triplet loss.  Or also "In the visual image domain, the positives are augmentations (transformations) of the same image while the negatives are augmentations from others", which is not necessarily true. This also occurs in the literature background. Simply making less specific claims or by allowing for the existence of other alternatives (e.g., stating "may" or "usually") could solve the issue. Another is "The feature-generating model is frozen while a single neuron is trained to predict each label", which is misleading since neurons are computational nodes, it is the linear layer parameters being trained.

Certain methods, such as Song et al., 2024 and 3KG, should likely be referenced in the literature background in greater detail. There is some methodology mixed into the results section (Song et al., 2024), which may or may not be necessary, but can hurt the flow.

There are typos ("same train/test splits..") and a random section link ("Unique patients1").

Calling the dataset-specific training and evaluation "the first investigation of its kind" is an ambitious statement, where the abstract had me expecting some kind of domain shift/adaptation methodology to be applied. The analysis seems good, does emphasize their multi-source dataset, and is important to understand how data distributions affect performance (e.g., healthy versus unhealthy subjects); However, the results of this analysis left me unsure as to what explicit value it added to this particular study.

More discussion is warranted as to why performance on the fully supervised benchmarks would become similar after a finetuning dataset of 100k is reached.

The paper could have benefited from additional non-tabular figures, such as a data scaling graph.

**Questions:**

Is it fitting for this paper to introduce a foundation model given the nature of the experimentation performed, as well as those related points raised under "Weaknesses"?

I started (and sort of left) the paper wondering: Is TA-PCLR a combination of individual existing methods which form a novel aggregate approach? Even by name, it is clear that TA-PCLR builds off the PCLR method, and yet how is never made totally clear - one would have to read the PCLR paper to learn this. Is it simply adding random temporal cropping to the PCLR method? Was the method/cropping approach inspired by other work in the ECG or timeseries space?

Considering, "The number of ECGs from each patient greatly differs and thus the training epoch is defined as one complete iteration for all unique patients with the positive views randomly sampled at training time. In this way, the training is not biased by patients having more ECGs while still exploiting the available data diversity." I agree that this maintains better overall population diversity, however I would be curious to know whether this would lead to biasing the model away from representing the conditions of those people with chronic health issues (i.e., the people who do have those higher ECG counts). What if it were those people whom were the most important to represent well?

---

> ### Author Response · Authors · 2024-11-25
> **Response 1 to Reviewer QAYi**
>
> We wish to thank you for your detailed and insightful comments. We tried our best to respond to each comment in the Weaknesses and Questions sections of your review. We have updated our paper in light of your reviews and would be thankful for your time to go through our response and revised manuscript. We hope that you will reconsider your previous score.
>
> **Reviewer comment:**
> This paper introduces a novel method more than it does a foundation model. Although they do pretrain on a large dataset, there is a lack of experimentation which only seems reasonable when releasing a foundation model. Namely, model scaling, considering how they adopt a relatively smaller architecture; They pride themselves on this point, however, this is unfounded due to a lack of confirmation that such a small deep neural network could make the most effective use of the vast pretraining dataset. This would have been vital to confirm.
>
> **Response:**
> You have pointed out an interesting research question that will be important for future research. In the scope of our current work, we select the particular Resnet architecture with ten convolutional layers (about six million parameters) based on the fact that this architecture has been extensively explored in the previous literature. Using a similar backbone we can fairly demonstrate the efficacy of our approach: our pre-training strategy and then the large unlabelled, diverse cohort. Prior research recommends the rule of ten times more data, as model parameters (https://www.sciencedirect.com/science/article/pii/S1755534518300058) so the size of the model with more than 5 million parameters is still quite large for our dataset of six million. Considering that the model performs equally well or even better at tasks, compared to supervised approaches, can be an indication that the representations - and hence the size of the model - are adequate to effectively capture ECGs. We present the model as a foundation model that can greatly facilitate the performance of any generic ECG-based supervised task, as the availability of labeled datasets is scarce in the medical domain. The long training times and resource constraints prevent us from exploring architecture scalability in the scope of our current work but we highlight the importance of this investigation in our conclusions in Section 5.
>
> **Reviewer comment:**
> The literature background refers to just a couple existing foundation models (and only in passing).
>
> **Response:**
> We add a paragraph about foundation models in Section 2.
>
> **Reviewer comment:**
> There is no interpretability method.
>
> **Response:**
> Interpretability is an important consideration for medical data and we add some results regarding interpretability in Appendix C. Figures 3 and 4 represent the t-SNE representation of the model embedding, and Figure 5 shows the importance of the different ECG segments using Grad-CAM. We also highlight the importance of future research in conclusions.
>
> **Reviewer comment:**
> They also report just one metric per task, which greatly reduces study comparability. Considering these points, it seems lacking for a foundation model paper, even though it seems sufficient for simply introducing a novel pretraining method.
>
> **Response:**
> The macro AUC has been recommended as a threshold-free metric for result comparison and most frequently employed in the past work thus being essential for comparability. The precision, recall, accuracy, and F1 greatly depend on the threshold used and thus may provide inconsistent information. We add these metrics for the PTB-XL super and sub classes in Appendix B Table 8 with a threshold of 0.5. We would like to mention that the values will be greatly improved by optimizing the threshold.
>
> **Reviewer comment:**
> Considering this paper is introducing a new pretraining method, it may have benefited from an ablation study of its various pretraining components beyond comparison to PCLR. Or, if the authors believe that this is all that's necessary as an ablation, then explain this in more detail to better communicate their contribution.
>
> **Response:**
> Thanks for pointing out the weakness in reporting the results. The main components of the approach can be regarded as patient contrastive augmentation, temporal augmentation, and the large pre-training dataset, and Table 2 represents an ablation study of these components. We have removed results for the PCLR using MGH and restructured the format so the configurations can be clearer for the readers. The table demonstrates that each component is essential to the success of the approach. To further clarify we restructure section 4.1 adding paragraph titles and rephrasing.

---

> > ### Author Response · Authors · 2024-11-25
> > **Response 2 to Reviewer QAYi**
> >
> > **Reviewer comment:**
> > Certain methodological matters are somewhat ambiguous. It was unclear whether hyperparameter tuning was performed. It is not made abundantly clear that the positive pairs are necessarily augmented instances of the same recording (and not from different recordings of the same patient). It is unclear whether there was any signal preprocessing (such as standardization), and if not, this may be worth stating.
> >
> > **Response:**
> > We apologize that the text did not clarify. A detailed hyperparameter tuning was not performed but the learning rate was optimized, from a range of 0.1 to 0.00001. The positive pairs are two different ECGs from the same patient taken at different times. Apart from the bandpass filtering mentioned no other preprocessing of the data was performed. Although the experiments conducted in Section 4.1 involved standardization of the ECG features for the supervised training.
> >
> > **Reviewer comment:**
> > Certain purported claims are left unsubstantiated, for example: "Human perception is limited by low visual accuracy, gaps in theoretical knowledge, and the complexity of the diverse, non-linear, interrelations."
> >
> > **Response:**
> > These observations have been made in past literature (https://www.nature.com/articles/s41597-023-02153-8). We have added the reference in the paper.
> >
> > **Reviewer comment:**
> > Certain references may be misleading provided its context, for example, "Hybrid techniques combining contrastive learning and generative pretraining based on transformer architecture have been implemented to train foundation models for ECG feature extraction (Song et al., 2024; McKeen et al., 2024)" may imply that ECG-FM (McKeen et al., 2024) using generative pretraining, which is not true.
> >
> > **Response:**
> > Thanks we have corrected the mistake as you rightly pointed out.
> >
> > **Reviewer comment:**
> > Certain definitions are questionable, for example, the statement that, "There are diverse contrastive learning approaches, differing in their definitions of the positive and negative instances and the loss computations" negates how certain contrastive methods do not apply here, such as using a triplet loss.
> >
> > **Response:**
> > We have rephrased using “mostly”.
> >
> > **Reviewer comment:**
> > Or also "In the visual image domain, the positives are augmentations (transformations) of the same image while the negatives are augmentations from others", which is not necessarily true.
> >
> > **Response:**
> > Rephrased inserting “usually” .
> >
> > **Reviewer comment:**
> > This also occurs in the literature background. Simply making less specific claims or by allowing for the existence of other alternatives (e.g., stating "may" or "usually") could solve the issue. Another is "The feature-generating model is frozen while a single neuron is trained to predict each label", which is misleading since neurons are computational nodes, it is the linear layer parameters being trained.
> >
> > **Response:**
> > The neuron is the computational node in a linear layer but here the neuron is mentioned to emphasize that no hidden layers are used and the linear layer consists of one neuron for each target label.
> >
> > **Reviewer comment:**
> > Certain methods, such as Song et al., 2024 and 3KG, should likely be referenced in the literature background in greater detail. There is some methodology mixed into the results section (Song et al., 2024), which may or may not be necessary, but can hurt the flow.
> >
> > **Response:**
> > The description of Song et al., 2024, etc is removed from the results section. Some details are added in Section 2 concerning these approaches but space constraints do not allow more detailed descriptions.
> >
> > **Reviewer comment:**
> > There are typos ("same train/test splits..") and a random section link ("Unique patients1").
> >
> > **Response:**
> > Corrected
> >
> > **Reviewer comment:**
> > Calling the dataset-specific training and evaluation "the first investigation of its kind" is an ambitious statement, where the abstract had me expecting some kind of domain shift/adaptation methodology to be applied. The analysis seems good, does emphasize their multi-source dataset, and is important to understand how data distributions affect performance (e.g., healthy versus unhealthy subjects); However, the results of this analysis left me unsure as to what explicit value it added to this particular study.
> >
> > **Response:**
> > We rephrase the abstract removing "the first investigation of its kind". The analysis is highly pertinent to the particular domain as in literature for ECG classification, often labe-specific comparisons are performed that may not be meaningful. We also wish to stress that our multi-centered dataset improves a generalization of the approach for diverse data distributions.

---

> > > ### Author Response · Authors · 2024-11-25
> > > **Response 3 to Reviewer QAYi**
> > >
> > > **Reviewer comment:**
> > > More discussion is warranted as to why performance on the fully supervised benchmarks would become similar after a finetuning dataset of 100k is reached.
> > >
> > > **Response:**
> > > The performance in this case saturates at around 200k since the test is similar to a linear probe but using an MLP head, to optimize training times in the scope of current work. We wished to highlight the performance in the low-data paradigm for the simple setup but subsequent tests for the external cohort in Table 4 demonstrate superior performance to SOTA supervised learning with fine-tuning the feature-generating backbone. Additional results in Appendix B Table 7 provide a wider comparison.
> > >
> > > **Reviewer comment:**
> > > The paper could have benefited from additional non-tabular figures, such as a data scaling graph.
> > >
> > > **Response:**
> > > Thanks, we have replaced the previous Table 3 with Figure 2.
> > >
> > > **Reviewer comment:**
> > > Is it fitting for this paper to introduce a foundation model given the nature of the experimentation performed, as well as those related points raised under "Weaknesses"?
> > >
> > > **Response:**
> > > We have responded to the previous comments. A foundation model is defined by generalization achieved through training on large unlabelled data. The greatly superior performance of our model for a range of different labels, achieved through our pretraining approach and large cohort, positions our model as a foundation model that can be exploited for any generic task. We have added the additional results requested in the appendix.
> > >
> > > **Reviewer comment:**
> > > I started (and sort of left) the paper wondering: Is TA-PCLR a combination of individual existing methods which form a novel aggregate approach? Even by name, it is clear that TA-PCLR builds off the PCLR method, and yet how is never made totally clear - one would have to read the PCLR paper to learn this. Is it simply adding random temporal cropping to the PCLR method? Was the method/cropping approach inspired by other work in the ECG or time-series space?
> > >
> > > **Response:**
> > > You are correct that the TAPCLR is a combination of existing patient contrastive and temporal augmentations that we have cited. We have tried to further clarify specifically in Section 3.3 to
> > > explain the choice and details about the augmentations. We select these particular augmentations as being harmless to the physiological characteristics of an ECG. The temporal augmentations included random cropping and zero masking of 20% of the signal.
> > >
> > > **Reviewer comment:**
> > > Considering, "The number of ECGs from each patient greatly differs and thus the training epoch is defined as one complete iteration for all unique patients with the positive views randomly sampled at training time. In this way, the training is not biased by patients having more ECGs while still exploiting the available data diversity." I agree that this maintains better overall population diversity, however, I would be curious to know whether this would lead to biasing the model away from representing the conditions of those people with chronic health issues (i.e., the people who do have those higher ECG counts). What if it were those people who were the most important to represent well?
> > >
> > > **Response:**
> > > The particular strategy not only balances the training for different subjects but also exploits the high number of ECGs when available. Each epoch uses a different pair thus utilizing all the data diversity for learning the representations. The fact that the pretraining performs best for the BIDMC with most ECGs per patient demonstrates that the training is able to benefit from these multiple ECGs.

---

### Meta-Review · Area_Chair_KRSN · 2024-12-19

**Metareview:**

This paper proposes temporally augmented patient contrastive learning for ECG data. The idea is to randomly crop ECG data as a form of augmentation with minimal changes to the underlying clinical pathology implied by the cropped ECG signal. This is then combined with an existing approach for patient-centered contrastive learning to build representations of ECG data. The experiments are run on a large, pretraining dataset and the representations used to make predictions of age, sex, mortality etc. I think the biggest issue that came out of the reviews and rebuttal period was around novelty -- I think the specific choice of augmentation used here is novel but most of the reviewers felt this alone was not sufficient. There were issues around writing, ablations and experiments that I think were well responded to and addressed during the rebuttal phase by the authors. One suggestion to expand the contributions by the work is to study (a) whether this form of augmentation works across different training and neural network architectural choices  and (b) better update their proposal with improved graphics and visual aids on why their method works.

**Additional Comments On Reviewer Discussion:**

The authors do a good job providing detailed explanations and further clarifications about their work. However, at the conclusion of the discussion period, the reviewer felt that a single proposal around a form of augmentation was insufficient depth in terms of contribution.

---

### Decision · Program_Chairs · 2025-01-22

Reject